# Targeting SUMO2 reverses aberrant epigenetic rewiring driven by SS18::SSX fusion oncoproteins and impairs sarcomagenesis

Rema Iyer[1], Anagha Deshpande[1], Aditi Pedgaonkar[1], Pramod Akula Bala[2], Taehee Kim [3], Gerard L Brien[4,5], Darren Finlay [6], Kristiina Vuori [6], Alice Soragni [3], Hiromi I Wetterstein[7], Rabi Murad[2] & Aniruddha J Deshpande [1]✉

## Abstract

Synovial sarcoma (SySa) is an aggressive soft tissue sarcoma with an urgent need to develop targeted therapies. Here, we exploited specific vulnerabilities created by transcriptional rewiring by the fusion protein SS18::SSX, the sole oncogenic driver in SySa. To uncover genes that are selectively essential for the fitness of SySa cells compared to other tumor cell lines, we mined the Cancer-Dependency-Map data. Targeted CRISPR library screening of SySa-selective candidates revealed that the small ubiquitin-like modifier 2 (SUMO2) constituted one of the strongest dependencies both in vitro and in vivo. TAK-981, a clinical-stage small-molecule SUMO2 inhibitor potently suppressed growth and colony-forming ability. Transcriptomic profiling showed that SUMO2 inhibition elicited a profound reversal of the gene expression program orchestrated by SS18::SSX fusion. Further, genetic depletion or SUMO2 inhibition reduced global expression levels and chromatin occupancy of the SS18::SSX fusion protein with a concomitant reduction in histone 2A lysine 119 ubiquitination (H2AK119ub), an epigenetic mark facilitating SySa pathogenesis. Taken together, our study identifies SUMO2 as a novel, selective vulnerability in synovial sarcoma, suggesting new avenues for targeted treatment of soft tissue tumors.

**Keywords** SUMO2; Synovial Sarcoma
**Subject Categories** Cancer; Chromatin, Transcription & Genomics; Post-translational Modifications & Proteolysis

## Introduction

Synovial sarcoma (SySa) belongs to a subcategory of sarcomas called soft-tissue sarcomas which accounts for 5–10% of all soft-

tissue tumors (Mastrangelo et al, 2012) and is more prevalent in adolescents and young adults (Moch, 2020). Approximately 30% of SySa cases occur in patients under 20 years of age (Speth et al, 2011; Sultan et al, 2009). This disease is characterized by an oncogenic fusion protein SS18::SSX formed by the translocation of (X;18) (p11.2;q11.2) (Ladanyi, 1995; Sorensen and Triche, 1996), which leads to the fusion of the SS18 gene to one of three SSX genes (SSX1, SSX2 or rarely to SSX4) on chromosome X. Although the SS18::SSX fusion has been characterized for more than three decades, therapies that target this fusion, or the oncogenic program driven by these fusion proteins remain to be identified.

The SS18::SSX fusion protein interacts with the SWI/SNF (BAF) complex, a large, chromatin modifying complex dysregulated in many human cancers. This interaction displaces the full-length SS18 as well as the SMARCB1/BAF47 protein from the BAF complex, altering its normal composition and function (McBride et al, 2018). The modified BAF complex then colocalizes with the Polycomb Repressive Complex 2 (PRC2) (Weber et al, 2021), leading to dysregulated transcriptional changes that are important for the oncogenesis of synovial sarcoma. This aberrant interplay between the BAF and PRC complexes results in the upregulation of several oncogenic pathways, including the Wnt/β-catenin (Barham et al, 2013; Ng et al, 2005), FGFR (DeSalvo et al, 2021), and NOTCH (Rota et al, 2012) pathways, while downregulating tumor suppressors such as EGR1 (Ciarapica et al, 2011; Su et al, 2010) and copy number variations of CDKN2A (CDKN2A) Gene Deletion Is a Frequent Genetic Event in Synovial Sarcomas) to name a few.

SS18 in the fusion is part of the canonical BAF complex and is associated with transcriptional activation. However, the SSX portion of the fusion protein is known to be repressive in function and binds regions rich in H2AK119ub1 (McBride et al, 2021), deposited by the non-canonical PRC1.1 complex. Although the SSX portion does not contain a direct ubiquitin binding site, recent findings indicate that it specifically binds to H2AK119ub-decorated sites via the 'H3-H2AK119ub' basic groove (Benabdallah et al,

[1]Cancer Genome and Epigenetics Program, National Cancer Institute-Designated Cancer Center, Sanford Burnham Prebys Medical Discovery Institute, La Jolla, CA 92037, USA. [2]Computational Biology Core, Sanford Burnham Prebys Medical Discovery Institute, La Jolla, CA 92037, USA. [3]Department of Orthopedic Surgery, David Geffen School of Medicine, University of California, Los Angeles, Los Angeles, CA 90095, USA. [4]Cancer Research UK Edinburgh Centre, Institute of Genetics and Cancer University of Edinburgh, Edinburgh, UK. [5]MRC Human Genetics Unit, Institute of Genetics and Cancer, The University of Edinburgh, Edinburgh, UK. [6]Cancer Molecular Therapeutics Program, National Cancer Institute-Designated Cancer Center, Sanford Burnham Prebys Medical Discovery Institute, La Jolla, CA 92037, USA. [7]Department of Pathology, Moores Cancer Center, and Sanford Consortium for Regenerative Medicine at the University of California, San Diego, La Jolla, CA, USA. ✉E-mail: adeshpande@sbpdiscovery.org

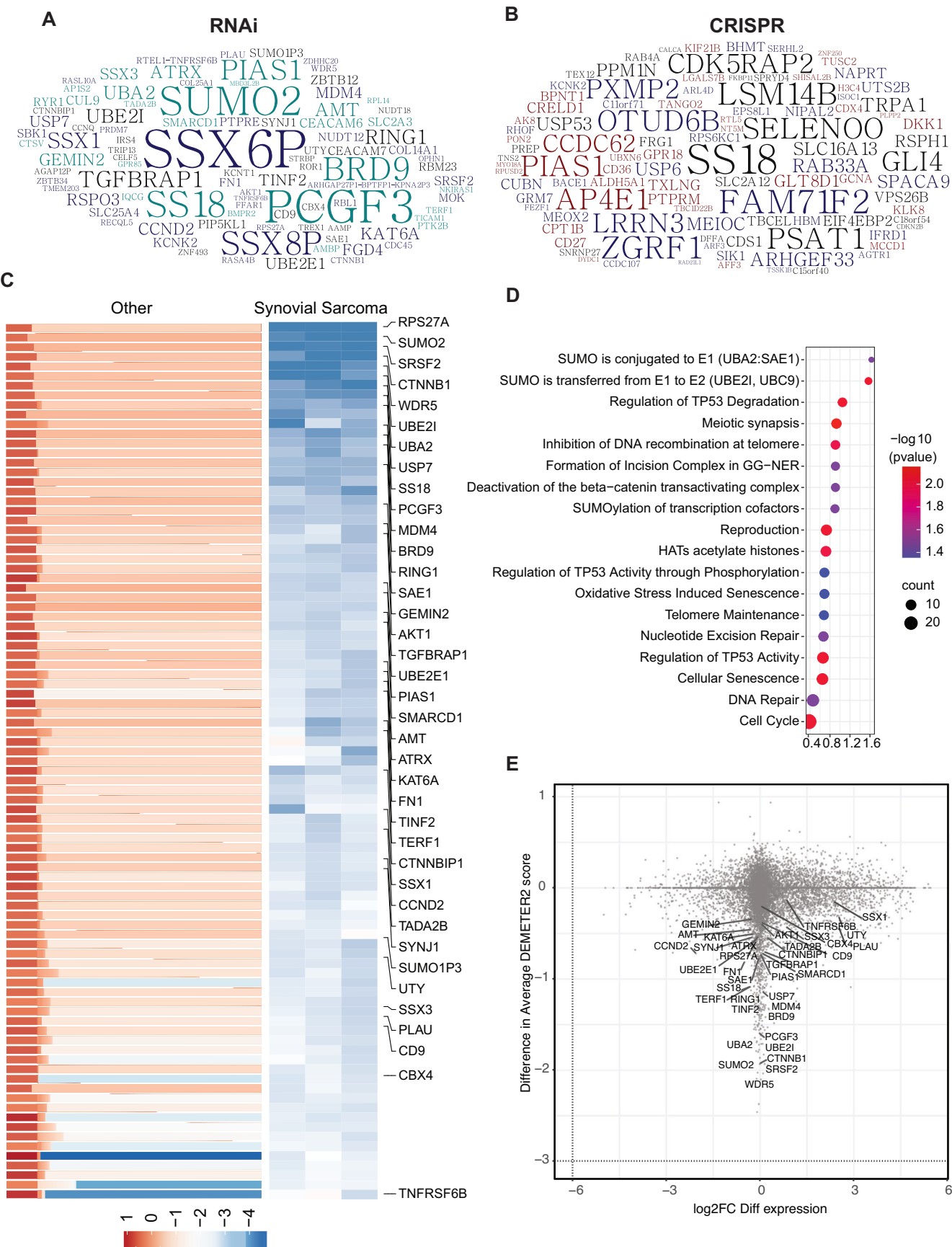

**Figure 1. Synovial sarcoma dependencies identified through DepMap screens.**

(A, B) Top 200 genes identified as selective dependencies using T-statistic scores of (A) RNAi data or (B) CRISPR screening data are shown in the word cloud. Font size is proportional to the negative log 10 adjusted $P$ value with a larger font indicating a higher dependency of the gene in synovial sarcoma cell lines compared to all other cell lines in the DepMap database. (C) Heatmaps representing DEMETER2 scores, as a quantitative dependency metric of human synovial sarcoma cell lines to each gene in RNAi screens. Relative DEMETER2 scores for non-synovial sarcoma cell lines (left) compared to synovial sarcoma cell lines (right) are depicted for the top differentially essential genes. (D) Bubble plot showing the top significantly enriched pathways (adjusted $P$ value < 0.05) from the REACTOME database, based on genes identified as top dependencies in synovial sarcoma. $P$ values were calculated using the Over-Representation Test (ORT) implemented in the MAGeCK pipeline's GSEA Reactome module. (E) Scatter plot showing the relationship between gene dependency (measured by the difference in average DEMETER2 scores) and differential transcript expression (log2 fold change FC differential expression) for all genes, with key synovial sarcoma selective essential genes labeled.

2023). This abnormal interaction leads to the unraveling of the nucleosome, redirecting the BAF complex to regions of chromatin occupied by the polycomb complex, which is one of the key mechanisms responsible for the epigenetic rewiring that drives synovial sarcoma pathogenesis.

Given the lack of targeted treatments in synovial sarcoma, a systematic approach to identify clinically tractable dependencies may yield valuable new candidates for therapy. Functional genomic approaches such as RNAi and CRISPR–Cas9 screens are powerful tools for forward genetics and have been effectively employed for the unbiased discovery of factors important for the viability of cancer cells (Hart et al, 2015; GTEx Consortium, 2020; Doench et al, 2014; Dempster et al, 2019). These large-scale screens can be used to identify vulnerabilities that are selectively essential for certain mutational subtypes (such as BRAF or KRAS mutated cancers) (Jung et al, 2021), or to nominate candidate targets selectively required for cancer types of interest (Behan et al, 2019). In this study, through an analysis of the DepMap RNAi and CRISPR datasets, we identified genes that are selectively essential in SySa cell lines compared to other cancer cell lines. Custom pooled screens of the top SySa selective vulnerabilities revealed the small ubiquitin-like modifier 2 (SUMO2) as one of the most significant dependencies both in vitro as well as in vivo. Importantly, small molecule inhibition of SUMO2 using TAK-981, a mechanism-based inhibitor of the SUMO2-activating enzyme (SAE) specifically led to a diminution of the fusion protein expression, chromatin occupancy, concomitant reversal of the genetic and epigenetic "lesions" characteristic of the SySa fusion proteins and strongly impaired SySa pathogenesis in vitro and in vivo. Taken together, our results reveal SUMO2 inhibition as an attractive therapeutic strategy in synovial sarcoma.

## Results

### Analysis of functional genomic screens identifies novel and known genetic vulnerabilities in synovial sarcoma

To identify potential genetic dependencies selective to synovial sarcoma, we analyzed gene dependency data from DepMap RNAi as well as CRISPR-Cas9 screen datasets and selected genes that have a higher essentiality in SySa compared to other cell lines (Fig. 1A–C). The list of SySa-selective dependencies identified through this analysis included SS18 and SSX genes that constitute the pathogenic fusions in SySA, as well as targets that have been proposed and validated by other groups including PCGF3 and BRD9 (Brien et al, 2018) (Fig. 1A–C). Our analysis also revealed several candidate SySa-selective genes that have not hitherto been studied in the context of SySa pathogenesis

(Fig. 1A–C; Dataset EV1). From the synovial sarcoma cell lines represented in the DepMap database, we selected top 200 genes from each of the datasets based on their DEMETER2 (RNAi) and Chronos (CRISPR) scores. From these lists, 351 unique genes were selected (Dataset EV1). We then conducted pathway analysis using Enrichr (Chen et al, 2013) to identify potential enrichment for biological pathways in the SySa-selective dataset. This analysis revealed that there was a striking enrichment for the SUMO conjugation and SUMO transfer Reactome pathway (adjusted $p$ values of 0.03 and 0.009, respectively) and multiple members of the sumoylation machinery appeared as hits in the SySa-selective dependencies dataset including UBA2, SAE1, UBE2I, SUMO2 and PIAS1 (Table EV1; Fig. 1D).

Other biological pathways enriched in this SySa-selective dependency data included genes involved in meiotic synapse formation, deactivation of the beta−catenin transactivating complex, histone acetylation, and regulation of p53 activity. Analysis of these SySa-selective dependencies using the STRING database showed enrichment in protein complexes involved in chromosome organization, WNT signaling, BAF complex, and PRC1 activity (Appendix Fig. S1) which are known dependencies in synovial sarcoma (Brien et al, 2018; Michel et al, 2018; Benabdallah et al, 2023; McBride et al, 2018). Novel biological pathways and protein complexes identified included the SUMO2-UBE2I complex, the SAGA, and the synaptojanin complex (Appendix Fig. S1). Next, we wanted to evaluate whether genes selectively essential for SySa were differentially expressed at the transcriptional level in SySa cell lines compared to other cancer cell lines. Thus, we calculated the fold change for each of these genes between SySa and non-SySa cancer cell lines in the Cancer Cell Line Encyclopedia (CCLE) and plotted it against the relative dependency values (DEMETER2) (Fig. 1E). In this analysis, we observed that while genes such as SSX1 and SSX3 were indeed much more highly expressed in SySa compared to non-SySa cell lines, genes such as BRD9, PCGF3, and SUMO2 had no noticeable difference in expression between these cell lines (Fig. 1E). This analysis indicates that while the relatively higher dependency of SySa cell lines on the SSX genes may result from their higher expression in cell lines from this lineage compared to others, the dependence on genes such BRD9, PCGF3, and SUMO2 may instead be explained by a relatively higher activity of these proteins in SySa compared to other cancers.

### In vivo and in vitro CRISPR screens nominate new candidate targets in synovial sarcoma

Building on our previous analysis, we sought to test these SySa-selective dependencies more comprehensively and investigate their essentiality in an in vitro as well as in vivo setting. To do so, we set up pooled CRISPR/Cas9 screens for the SySa-selective genes. First, we assessed the activity of Cas9 in HS-SY-II cells expressing Cas9 to

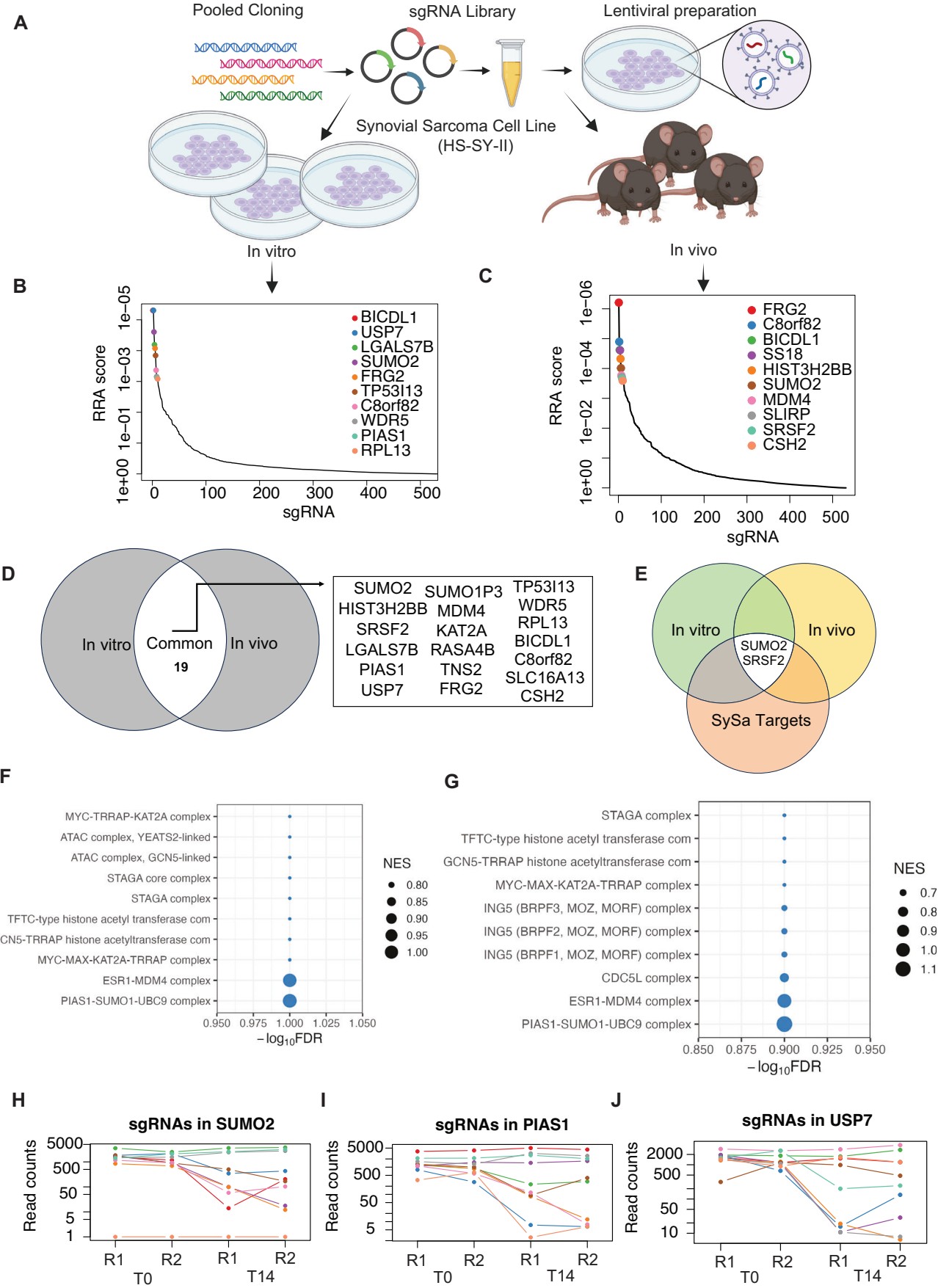

**Figure 2. In vivo and in vitro screening reveal top synovial sarcoma-selective dependencies.**

(A) Schematic representation of in vivo and in vitro pooled CRISPR screens in HS-SY-II cell line. (B, C) Analysis of pooled in vitro (left) and in vivo (right) CRISPR/Cas9 screens using the MAGeCK RRA algorithm. Plot shows the relationship between sgRNA of different genes as well as control non-targeting sgRNA (X axis) and their statistical significance (RRA score) (Y axis). The top ten significant genes are labeled. RRA is robust rank aggregation algorithm as assessed using MAGeCK. (D) A Venn diagram illustrating genes commonly essential in both the in vitro and in vivo pooled CRISPR/Cas9 screens. Common genes in the union are labeled. (E) A Venn diagram illustrating genes common to the in vitro and in vivo screens as well as in the core synovial sarcoma oncogenic program. Common genes in the union are labeled. (F, G) CORUM complex pathway enrichment of hits in the in vitro (F) and in vivo screens (G). −log10(FDR) or false discovery rate is shown on the X axis. The size of the bubbles indicates normalized enrichment scores of each pathway. Data analyzed using MAGeCK-MLE module. (H–J) Normalized read counts of multiple individual sgRNAs for SUMO2 (H), PIAS1 (I) and USP7 (J) showing the difference between T0 and T14 time points.

ensure high editing efficiency (indel percentage identified as ~92% and a knockout score of 90 using ICE (Conant et al, 2022)). With these optimized conditions, we then performed parallel in vivo and in vitro CRISPR screens (schematic Fig. 2A). HS-SY-II-Cas9 cells expressing Cas9 were transduced with the screening library in duplicate at a MOI of ~0.3. We then subcutaneously injected 2 million cells (~500× coverage) into the flanks of nude mice. In parallel, for the in vitro screen, we cultured the cells from each replicate for ~10 doubling times. There was strong replicate reproducibility for both the in vitro and in vivo results (Appendix Fig. S2). sgRNA abundance and distribution were quantified using MAGeCK Robust Rank Aggregation algorithm (Li et al, 2014). In vitro and in vivo hits were generally well correlated (Appendix Fig. S2), with the identification of several overlapping hits including KAT2A, C8orf82, SUMO2, FRG2, BICDL1, and LGALS7B (Fig. 2B–D; Table EV2). We then turned our attention to targets that were previously not described as dependencies of synovial sarcoma and ranked highly in both the in vivo as well as in vitro screens (Fig. 2D). To ensure that among these genes, we prioritize those more directly relevant to the fusion-driven oncogenic program, we overlapped them with genes that are regulated by the SS18::SSX fusion oncoprotein (SS18::SSX fusion targets) as shown by Jerby-Arnon et al (Jerby-Arnon et al, 2021) (Fig. 2E) (Table EV2). Of the genes that are strongly depleted in our in vitro and in vivo screens and are activated by SS18::SSX fusions in SySa cells, we were particularly interested in SUMO2. SUMO2 was one of the most essential genes in the in vitro (RRA score 5.29E-06), as well as in the in vivo screen (RRA score 7.95E-05). Interestingly, SUMO2 has been shown to be transcriptionally activated by SS18::SSX fusions in two independent SySa cell lines in prior studies(Jerby-Arnon et al, 2021). Pathway enrichment analysis showed that the top hits were enriched for proteins involved in the SUMO complex in both in vitro as well as in vivo screens (Fig. 2F,G). Individual sgRNAs for SUMOylation pathway genes showed a dramatic drop in read counts (Fig. 2H–J) further validating SUMO2 as a top candidate hit in our screens. A small molecule inhibitor—TAK-981—that selectively inhibits SUMO2 is currently in phase 1/2 clinical trial for Non-Hodgkin lymphoma (NCT04074330) and phase 1b/2 for refractory multiple myeloma (NCT047760180). We therefore earmarked SUMO2 as a novel candidate and a therapeutic target for further evaluation.

## TAK-981, a small molecule SUMO2 inhibitor impairs the growth of synovial sarcoma cells

To systematically test the effect of SUMO2 inhibition on synovial sarcoma cell lines, we first evaluated the effect of TAK-981 on

proliferation in four different human synovial sarcoma cell lines (SYO1, HS-SY-II, 1273/99, Aska-SS) as well as the epithelial squamous cell lung cancer cell line (SK-MES-I) and human embryonic kidney 293T cells (HEK-293T). TAK-981 treatment diminished sumoylation (Appendix Fig. S3) and significantly reduced the proliferation of these cell lines in a concentration-dependent manner, showing half maximal effective concentration ($EC_{50}$) in the nanomolar range in a CellTiter-Glo assay, with the HS-SY-II cell line exhibiting the strongest inhibition (Fig. 3A). Generally, SySa cells lines showed a substantially higher sensitivity to TAK-981 than non-synovial sarcoma cell lines SK-MES-I or HEK293-T cells (Fig. 3A). To determine the effect of TAK-981 on apoptosis, we performed Annexin V staining—on TAK-981 treated and untreated cells. The proportion of early and late apoptotic cells significantly increased in SYO1 and HS-SY-II cells after 24–72 h of TAK-981 treatment compared to DMSO-treated cells (Fig. 3B,C; Appendix Fig. S4). Additionally, cell cycle analysis using propidium iodide indicated reduction in cells in the S-phase (Fig. 3D). We then performed cell viability assays on 2D and 3D cultures for SYO1 and Aska-SS cell lines to determine whether these culture conditions resist TAK-981 treatment. In these studies, TAK-981 treatment led to a progressive and marked decrease in viability as measured by CellTiter-Glo in both 2D as well as 3D cultures conducted over 2, 3, and 4 days (Fig. 3E). Colony-forming assays for the HS-SY-II, SYO1, and 12273/99 cell lines using a TAK-981 titration series also demonstrated a dramatic and dose-dependent reduction in colony formation (Fig. 3F,G). A genetic knockdown of SUMO2 in HS-SY-II as well as SYO1 cell lines using a Doxycycline-inducible system also show a similar effect (Fig. 3H,I).

## TAK-981 treatment impairs transcription of key oncogenic pathways in synovial sarcoma cell lines

To comprehensively interrogate the transcriptomic changes occurring in synovial sarcoma cells upon TAK-981 treatment, we treated HS-SY-II (harboring the SS18::SSX1 fusion) and SYO1 cells (harboring the SS18::SSX2 fusion) (Appendix Fig. S5) with DMSO or TAK-981 and performed bulk RNA sequencing. Common to both HS-SY-II and SYO1, a total of 1087 differentially expressed genes (DEGs) were detected using the threshold of |fold change| >2 and adjusted P value < 0.01, of which 844 genes are upregulated, and 243 genes were downregulated, respectively (Dataset EV2). Of these, key cancer-associated genes shown to be upregulated by the SySa fusion(Jerby-Arnon et al, 2021) were downregulated by TAK-981 treatment, including CDX2, HOXA10, SUZ12, TYMS, AURKB, (Fig. 4A) and HOXC10 and SMC2 (Fig. 4B). Concomitantly, genes upregulated by the SySa fusions were down-regulated by TAK-981 treatment including KLF4, GADD45B, CXCR4 and

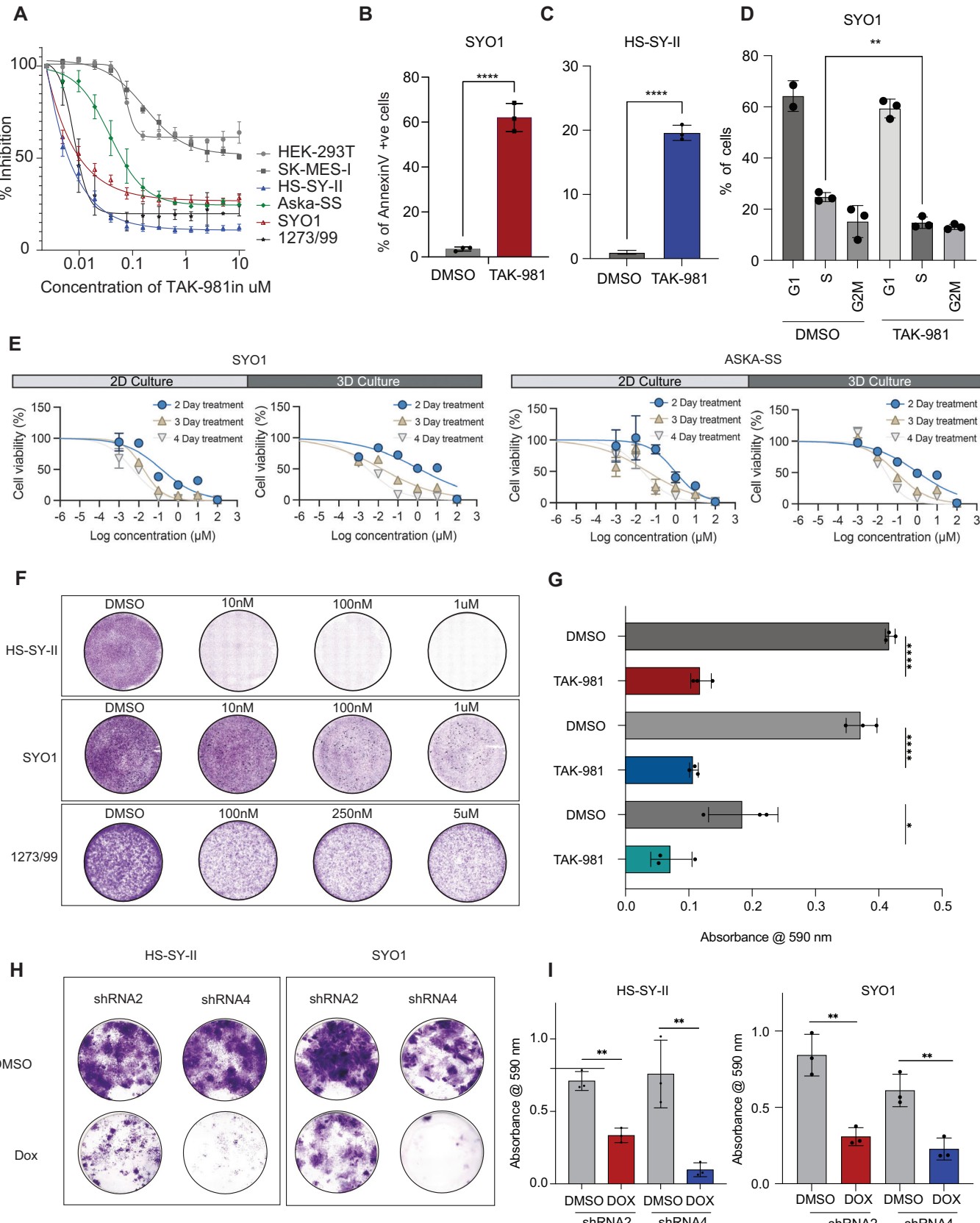

**Figure 3. Effect of TAK-981 on synovial sarcoma cell lines.**

(A) Viability of various SySa cell lines measured using Cell-Titer-Glo after 48 h of treatment with varying concentrations of TAK-981 is shown. X axis shows concentration of TAK-981 in μM and Y axis shows percent inhibition compared to vehicle-treated counterparts. N = 4 biological replicates. Error bars represent mean ± SD. (B, C) Percent of Annexin V positive SYO1 cells treated with 1 μM TAK-981 for 72 h. (B) and HS-SY-II cells treated with 25 nM TAK-981 for 24 h, P = 000012. (C) compared to their DMSO treated arms are plotted on the Y axis, N = 3 biological replicates. P values were calculated using a two-tailed Student's T test with ****P < 0.0001 (P = 0.00009). Error bars represent mean ± SD. (D) Bar graph representing the percent propidium iodide (PI) positive SYO1 cells in DMSO-treated compared to TAK-981-treated arm are shown (Y axis). Cells in G2, S and G2M phases are shown. N = 3 biological replicates, 24 h. post treatment. P values were calculated using a two-tailed Student's T test with **P < 0.01 (P = 0.00009). Error bars represent mean ± SD. (E) Percent viability (relative to DMSO-treated) SYO1 (left) or Aska-SS cells (right) 2, 3 or 4 days after treatment in 2D or 3D growth formats is plotted (Y axis). X axis shows conc. of TAK-981 in μM. N = 3 biological replicates. Error bars represent mean ± SD. (F) Representative images of crystal violet stained colonies for cell lines HS-SY-II, SYO1 or 1273/99 48 h after treatment with DMSO or varying concentrations of TAK-981. (G) Quantification of crystal violet stain intensity in TAK-981 (100 nM) treated cell lines when compared to DMSO. X axis shows absorbance at 590 nm Y axis is respective cell line. N = 3 biological replicates. P values were calculated using a two-tailed Student's T test with ****P value < 0.0001 (0.000009 for HS-SY-II and 0.000054 for SYO1), *P value < 0.05 (0.0366 for 1273/99). Error bars represent mean ± SD. (H) Images of crystal violet stained colonies of HS-SY-II and SYO1 cells transduced with SUMO2 shRNAs and treated with DMSO (top) or induced with Doxycycline (DOX) at 4 μg/ml concentration for 11 days. Representative images are shown. (I) Quantification of DMSO vs DOX for HS-SY-II (left) and SYO1 (right). Y axis shows absorbance at 590 nm for shRNA2 and shRNA4. N = 3 biological replicates. P values were calculated using a two-tailed Student's T test with **P value < 0.002 (HS-SY-II shRNA2 P = 0.0034, HS-SY-II shRNA4 P = 0.0067, SYO1 shRNA2 P = 0.0014 and SYO1 shRNA4 P = 0.0086). Error bars represent mean ± SD. Source data are available online for this figure.

*GDF15* (Fig. 4A,B). The commonly downregulated genes were highly enriched for cell cycle (adjusted P value 1.023e-39), cell cycle checkpoint (adjusted P value 2.000e-22) and S phase genes (adjusted P value 2.304e-17), and DNA replication-associated genes (adjusted P value 4.018e-13) in the Reactome database, consistent with cell cycle arrest following SUMO2 inhibition. Importantly, it has been shown that the high expression of cell cycle genes is a key feature of a subset of undifferentiated cells in synovial sarcoma patient samples and that these genes are regulated by SS18::SSX fusions. In our studies, these genes showed a significant downregulation upon TAK-981 treatment (Fig. 4C; Appendix Fig. S6A–D). Of genes that were commonly upregulated by TAK-981 treatment in the two cell lines, there was a significant enrichment of genes involved in collagen formation and extracellular matrix formation (adjusted P value 3.435e-8). Notably, TAK-981 treatment also led to the significant downregulation of several genes associated with resistance to doxorubicin, which is used in the treatment of synovial sarcoma(Barreto Coelho et al, 2021) (Fig. 4C,D) as assessed using gene set enrichment analysis (GSEA) (Subramanian et al, 2005). To further investigate this finding, we performed IC 50 synergy studies with Doxorubicin and TAK-981 using multiple cell lines. We find that HS-SY-II cells are most sensitive to Doxorubicin and TAK-981 with a mean Bliss score of 10.2 (Fig. 4E). Incidentally, YAMATO-SS and 1273/99 which are highly insensitive to Doxorubicin alone, showed sensitivity to Doxorubicin in the presence of TAK-981 (Appendix Fig. S7). These results support further evaluation of TAK-981 in combination with other chemotherapies.

## TAK-981 treatment specifically reverses the transcriptional signatures driven by SySa fusion proteins

Next, we investigated whether TAK-981 treatment specifically alters the expression of synovial sarcoma fusion target genes, as defined by Jerby-Arnon et al (Jerby-Arnon et al, 2021). In their study, the authors defined the SS18::SSX program by knocking down the SS18::SSX fusion and conducting a ChIP-seq analysis for the fusion. This allowed them to identify genes that were bound by the SS18::SSX fusion protein and whose expression was modulated by the knockdown of the fusion as direct targets and genes not bound by the fusion but modulated by its knockdown as indirect targets. We utilized this list of genes for a custom gene set enrichment analysis in the TAK-981 treated RNA-seq dataset.

In these analyses, we observed that in both SYO1, and HS-SY-II cell lines, TAK-981 treatment led to a dramatic reversal of the SS18::SSX-driven transcriptomic program. Specifically, genes repressed by the chimeric SS18::SSX fusion protein showed a significant upregulation in expression upon TAK-981 treatment as assessed using GSEA (Fig. 5A,B), and this included the KLF4, TBX3 and CXCR4 genes (Fig. 5C). Concomitantly, genes activated by SS18::SSX in synovial sarcoma were repressed (Fig. 5D,E), including HOX genes HOXC6, HOXA10 as well as SRSF1 and TYMS (Fig. 5F). Of note, the fact that this was evident both in the SYO1 cell line expressing the SS18::SSX2 fusion protein as well as in the HS-SY-II cell line expressing the SS18::SSX1 fusion strongly indicates that SUMO2 is critical for the transcriptional activity of both types of distinct SS18::SSX fusion oncoproteins.

Our observation that SUMO2 inhibition reverses the oncogenic program driven by two distinct SS18::SSX fusion oncoproteins indicates that SUMO2 is a critical node in regulating the oncogenic activity of these chimeric oncoproteins. Given the specific reversal of the SySa fusion-driven program, we sought to test the intriguing hypothesis that SUMO2 regulates the SS18::SSX fusion protein itself. For this, we cloned shRNAs targeting SUMO2 into a tetracycline-inducible plasmid and expressed the shRNAs in HS-SYII and SYO1 cells. qPCR results validated the knockdown of SUMO2 transcript expression following doxycycline induction of the shRNAs (Appendix Fig. S8). Strikingly, SUMO2 knockdown showed a reduction in SS18::SSX protein levels (Fig. 5G; Appendix Fig. S9). These results could be replicated using a pharmacologic approach—TAK-981 treatment led to a reduction in the levels of SS18::SSX1 protein in the HS-SY-II cell line (Fig. 5H) and SS18::SSX2 fusion in the SYO1 cell lines (Fig. 5I) as well as in the 1273/99 cell line (Appendix Fig. S10). These results provide a striking demonstration that SUMO2 inhibition modulates the levels of oncogenic fusion proteins that drive sarcomagenesis in SySa.

## SUMO2 inhibition diminishes SS18::SSX chromatin occupancy and reverses fusion-driven aberrant epigenomic changes in SySa cells

Next, we sought to assess whether TAK-981 treatment affects the chromatin occupancy of the SS18::SSX fusion protein. For this, we

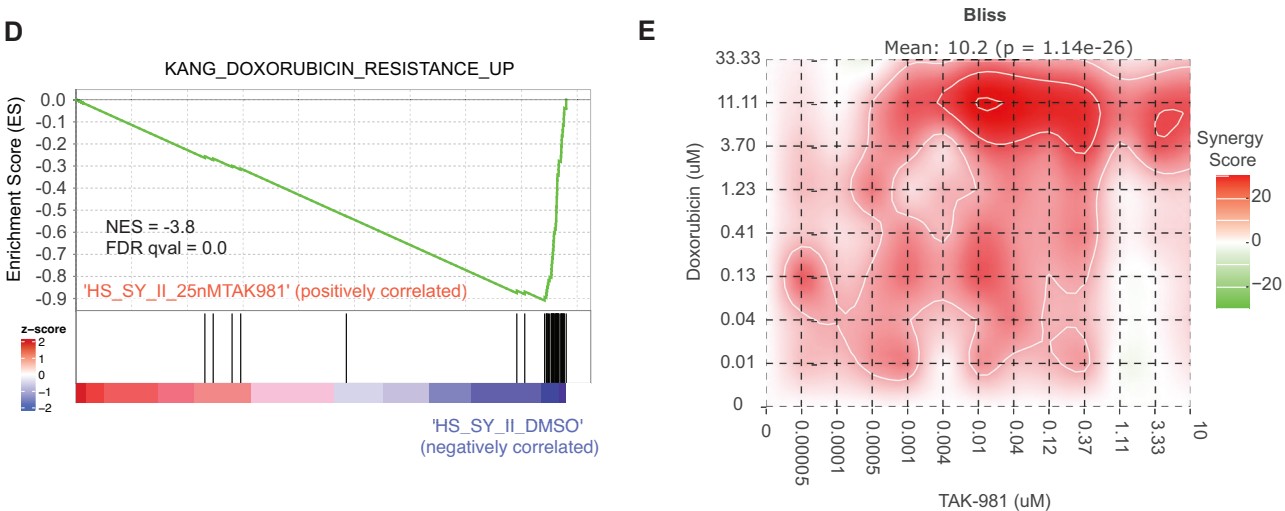

◄ **Figure 4.   Broad transcriptomic changes in TAK-981 treated synovial sarcoma cell lines.**

(A, B) Volcano plot illustrating differential gene expression in SYO1 (A) or HS-SY-II cells (B) treated with DMSO compared to TAK-981. $N = 2$ biological replicates, Each dot represents an individual gene. The $X$ axis represents log-2 fold change (DMSO Vs TAK-981 treated cells) and the $Y$ axis represents $-\log 10$ BH adjusted $P$ value. Red dots represent genes significantly upregulated in the TAK-981 treated compared to the DMSO treated arm with adjusted $P$ value < 0.05 and fold change >2. Blue dots represent genes that are significantly downregulated with $P$ value < 0.05 and fold change <0.5. Grey dots represent genes that are not significantly differentially expressed. Reported statistics derived using Wald Test from DESeq2 version 1.22.2. (C) The bar chart represents top 20 significantly enriched gene ontology (GO) terms of the C2: canonical pathways gene set. The horizontal axis represents pathways with positive (red) and negative (blue) normalized enrichment scores (NES). (D) GSEA analysis of the C2 Curated Datasets in MiSigDB for genes upregulated during doxorubicin resistance is shown for transcriptomic data of HS-SY-II cells treated with DMSO compared to TAK-981. NES Normalized Enrichment Score. (E) Dose response for the pairwise combination of TAK-981 (0–10 μM) and Doxorubicin (0 to 33 μM) in HS-SY-II cells after 48 h. of incubation. $P$ value for the average synergy score was derived using bootstrapping of the dose–response matrix and Bliss score was calculated using SynergyFinder.org.

performed Cleavage Under Targets & Release Using Nuclease (CUT&RUN) using the SS18::SSX-fusion specific antibody (see "Methods"). These studies demonstrated a substantial decrease in SS18::SSX2 fusion genomic occupancy as assessed using spike-in normalized CUT&RUN analysis in the TAK-981-treated compared to vehicle-treated arms (Fig. 6A). Specifically, TAK-981 treatment of SYO1 cells showed a 1.87-fold reduction in genome-wide chromatin binding signal of the SS18::SSX fusion compared to the DMSO treated cells, as computed from fraction of reads in peaks (FRiP) measured using consolidated peaks in DMSO replicates. A meta-analysis of the SS18::SSX binding signal at SySa target genes showed that this decrease was more prominent at the promoter proximal regions (Fig. 6B).

Since the increased H2AK119ub deposition has been linked to the pathogenic activity of the SS18::SSX fusions, we then sought to assess H2AK119 ubiquitination in TAK-981 treated cells. Chromatin immuno-precipitation (ChIP)-sequencing of H2AK119ub showed that concomitant with the loss of SS18::SSX expression, there was a substantial reduction in global H2AK119ub in TAK-981 compared to DMSO-treated cells (Fig. 6C). Specifically, there was a 1.53-fold reduction of genome-wide H2AK119ub levels in TAK-981 versus DMSO treated SYO1 cells, computed as fraction of reads in peaks (FRiP) measured using consolidated peaks in DMSO replicates (Fig. 6C). Immunoblotting studies further confirmed the loss of H2AK119ub in SYO1 cells with SUMO2 knockdown (Appendix Fig. S11) as well as in TAK-981 treated cells (Fig. 6D). A reduction in overall H2AK119ub was observed in other cell lines treated with TAK-981 as well (Appendix Fig. S12).

It has been previously noted that SS18::SSX fusions promote H2AK119ub. We therefore first analyzed the overall levels of H2AK119Ub on SySa target genes (Jerby-Arnon et al, 2021) compared to an equal number of expression-matched non-target H2AK119ub positive genes. This analysis in SYO1 cells demonstrated that SySa targets have a substantially higher overall H2AK119ub deposition than the well-controlled set of non-target genes (Fig. 6E). Furthermore, our analysis showed that TAK981 treatment led to a more substantial reduction in H2AK119ub at the SySa targets (Fig. 6F) than at the non-target genes (Fig. 6G).

Genes including the SS18::SSX-activated targets HOXA10 and SOX8 lost fusion occupancy, reduced H2AK119ub and decreased expression upon TAK-981 treatment (Fig. 6H). On the other hand, SS18::SSX repressed genes such as GADD45B, were upregulated in the presence of TAK-981, while showing a reduced fusion occupancy and H2AK119ub modification. These illustrative examples highlight the effect of TAK-981 on reversing the chromatin activity of the SS18::SSX fusion on both activated and repressed targets.

## TAK-981 impairs sarcomagenesis of SySa in vivo

To determine the antitumor activity of TAK-981 in vivo, we injected SYO1 (harboring the SS18::SSX2 fusion) or Aska-SS cells (harboring the SS18::SSX1 fusion) into the flanks of nude mice. When tumors became palpable, mice were treated with 25 mg/kg of TAK-981 or vehicle. A dosing schedule of 3 intraperitoneal injections a week for 5 weeks was maintained (Fig. 7A). Consistent with the in vitro assays, TAK-981-treated mice showed a remarkable reduction of tumor growth when compared to vehicle-treated mice. Tumor volumes in Aksa-SS injected mice were significantly reduced in the TAK-981-treated arm as were tumor weights (Fig. 7B–D). IHC analysis of the tumors stained with hematoxylin and eosin showed a marked reduction in the number of cells per unit area within TAK-981 treated tumors when compared to the vehicle-treated tumors (Fig. 7E–I), both in the periphery and center of the tumor (Fig. 7E,F). Ki67 staining revealed a ~60% decrease in Ki67 positivity in comparison with the vehicle-treated tumors indicating decreased proliferation (Fig. 7G–I). We observed that TAK-981 was well tolerated, and the mice maintained their body weight and showed no visible signs of toxicity through the dosing period (Appendix Fig. S13). Similar results were obtained for SYO1 injected mice, where TAK-981 treatment led to a significant decrease in tumor size (Fig. 7J–L), and a concomitant decrease in cellularity (Fig. 7M,N) and Ki67-positive cells (Fig. 7O–Q). The data demonstrates that TAK-981 efficiently inhibits tumor growth in SS18::SSX1 fusion containing ASKA-SS as well as SYO1 cell lines. Furthermore, on staining for the SS18::SSX fusion specifically obtained from in vivo tumor sections, we observed that TAK-981 treated tumors had a significant decrease in the percentage of cells with high SS18::SSX expression as assessed by immunohistochemistry compared to vehicle treated mice (Fig. 7R,S). Taken together, these data demonstrate that TAK-981 treatment has potent in vivo activity in in vivo models of SySa tumors.

## Discussion

Synovial sarcoma can be managed effectively with surgery and accompanying radiation therapy and/or chemotherapy in some patients - especially in children with localized disease. However, advanced stages of the disease present a much more difficult challenge and the prognosis in such cases remains poor. Developing more precise, targeted therapies for synovial sarcoma has been hampered by the lack of detailed understanding of the mechanisms that drive disease pathogenesis. The presence of the disease-defining SS18::SSX protein that is largely responsible for driving tumorigenesis has prompted several efforts in trying to understand the mechanistic underpinnings of this disease. Since SS18—the

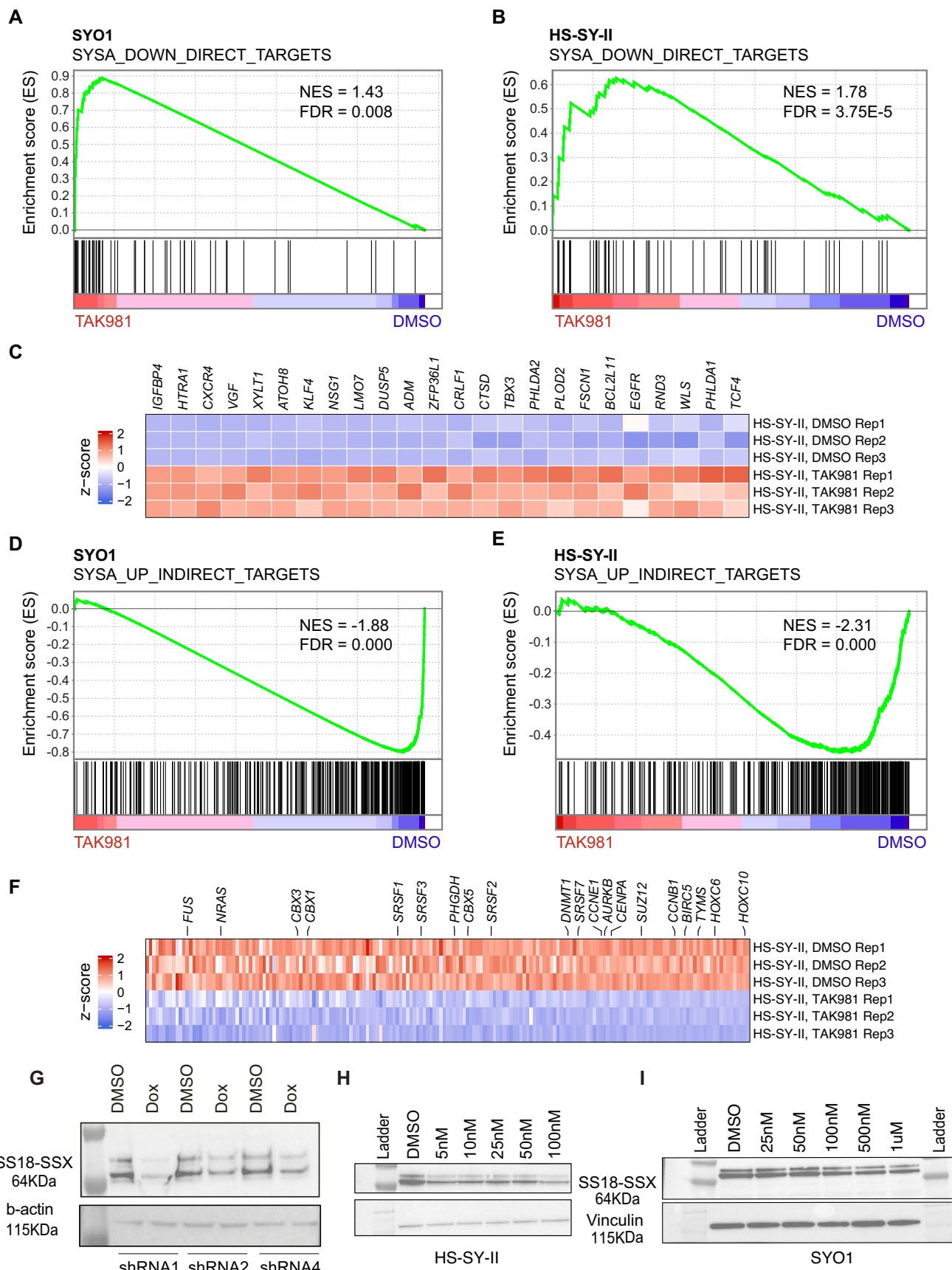

**Figure 5.  Treatment with TAK-981 leads to downregulation of oncogenic program in synovial sarcoma cell lines.**

(A, B) GSEA analysis of SS18::SSX fusion-repressed genes in SYO1 (A) or HS-SY-II (B) cells treated with TAK981 compared to DMSO is shown. Enrichment plots depict genes that are direct targets of the SS18::SSX fusion, which are upregulated upon SS18::SSX fusion knockdown (thus, SS18::SSX-repressed genes). Black vertical lines at the bottom indicate positions of individual genes in the set, with the green line representing the cumulative enrichment score ($Y$ axis). A positive normalized enrichment score (NES) indicates enrichment in the upregulated genes in SYO1 cells (A) and HS-SY-II cells (B). FDR $q$ values are indicated. (C) Heatmap displaying SS18::SSX fusion-repressed genes that are increased in TAK-981 treated compared to DMSO treated HS-SY-II cell line are shown. Select genes implicated in SySa pathogenesis are labeled. (D, E) GSEA analysis SS18::SSX fusion-activated genes in SYO1 (D) or HS-SY-II (E) cells treated with DMSO compared to TAK-981 is shown. Enrichment plots show genes that are indirectly activated by the SS18::SSX fusion and are thus downregulated upon SS18::SSX fusion knockdown. A negative NES indicates higher expression enrichment of these genes in the TAK-981 compared to DMSO arms in SYO1 (D) as well as HS-SY-II cells (E). (F) Heatmap displaying SS18::SSX fusion-activated genes that are reduced in TAK-981 treated compared to DMSO treated HS-SY-II cell line are shown. Select genes implicated in SySa pathogenesis are labeled. (G) Immunoblot analysis of whole-cell lysates from HS-SY-II cells stably expressing SUMO2 knockdown in a doxycycline-inducible system (shRNA1, 2, and 4), probed for the SS18::SSX1 fusion protein. Vinculin as a loading control is shown. (H, I) Immunoblot analysis of whole-cell lysates from HS-SY-II (H) or SYO1 cells (I) treated with varying denoted concentrations of TAK-981 and probed for the SS18::SSX1 fusion protein are shown. Vinculin is shown as a loading control. Source data are available online for this figure.

larger component of the SS18::SSX fusions—is a member of the BAF (aka SWI/SNF) chromatin remodeling complex, seminal studies sought to investigate how SS18::SSX fusions perturb normal BAF complex function. A series of studies showed that the SS18::SSX fusion protein replace the normal SS18 protein in the BAF complex, leading to the disruption of normal BAF complex activity (Kadoch and Crabtree, 2013) (McBride et al, 2018). Further studies demonstrated that this epigenetic rewiring fundamentally alters the chromatin crosstalk between the BAF complex and the polycomb regulatory complexes PRC1 and PRC2 (Boulay et al, 2021; Benabdallah et al, 2023). Specifically, studies showed that in addition to compromising normal BAF function, the SS18::SSX containing BAF complex evicts PRC2 from fusion bound sites (Boulay et al, 2021). Lastly, more recent studies have shown that the SS18::SSX fusions enhances the activity of the PRC1 (specifically the PRC1.1) complex, through stabilization of PCR1.1 core components, enhancing global H2K119ub (Benabdallah et al, 2023). In fact, a recent study showed elegantly, using a conditional mouse model of SySa driven by the SS18::SSX2 fusion protein, that the H2AK119ub mark is acquired gradually during tumorigenesis, ostensibly through the stabilization of key PRC1.1 complex members, enabling further fusion protein binding (Benabdallah et al, 2023). Since SS18::SSX fusions bind to H2AK119ub through the SSX reader domain that is retained in the fusion protein (Tong et al, 2024), a picture emerges where epigenetic rewiring by the SS18::SSX fusions drive a transcriptional feed-forward loop to sustain activity of the SySa oncotranscriptome (McBride et al, 2020; Benabdallah et al, 2023; Michel et al, 2018; Jerby-Arnon et al, 2021). Considering this, it is interesting to note that SUMO2 inhibition reverses this epigenetic rewiring by reducing levels of both the SS18::SSX fusion protein as well as of global and fusion-locus specific H2AK119 ubiquitination. These results indicate that SUMO2 is likely involved in key processes that sustain the transcriptional feed-forward loop characteristic of SySa tumor cells. The effectiveness of SUMO2 inhibition in synovial sarcoma models by specifically suppressing the pathogenic features of the SS18::SSX fusion oncoprotein indicate that SUMO2 is a highly selective vulnerability in synovial sarcoma as indicated by our analysis of the Dependency Maps (DepMap) data.

Recognizing the importance of SUMO2 in other malignancies, TAK-981—a specific SUMO2 inhibitor—has been developed for clinical testing for many solid tumors as well as hematological malignancies. Proof of concept of its efficacy has been shown in AML (Kim et al, 2023) and pancreatic cancer (Kumar et al, 2022) in

preclinical studies. However, its potential benefits in synovial sarcoma (SySa) have not been explored, and merit clinical investigation based on our findings.

Of note, SUMO2 inhibition using TAK-981 was recently shown to potentiate the antitumor immune response by activating CD8+ T-cells through modulation of type I interferon signaling (Lightcap et al, 2021). In this study, TAK-981 improved the survival of mice in models of colorectal cancer, enhancing the response of anti-PD1 or anti-CTLA4 antibodies. In future studies, it will be interesting to determine whether TAK-981 treatment has similar effects on augmenting antitumor immunity in synovial sarcoma in addition to the strong cell-intrinsic anti-oncogenic activity observed in our studies.

Importantly, our results showing that SUMO2 inhibition is effective in cells driven by different SySa fusions, irrespective of their carboxy-terminal fusion partner (SSX1 or SSX2) indicate that these inhibitors may be broadly applicable for SySa patients with distinct SySa fusion proteins. Also, since targeted therapies are more likely to be successful in combination with other cytotoxic agents, SUMO2 inhibitors may work more effectively in combination with currently used chemotherapies. Taken together, our results highlight the potential of SUMO2 inhibitors as promising therapeutic targets for SySa, with TAK-981 emerging as a particularly strong candidate for clinical testing in patients with synovial sarcoma.

## Methods

**Reagents and tools table**

| Reagent/resource | Reference or source | Identifier or catalog number |
| --- | --- | --- |
| **Experimental models** | | |
| Nu/Nu | Jackson Laboratory | 007850 |
| HEK-293T | Takara Bio | 632180 |
| SYO1 | | |
| HS-SY-II | Riken | |
| Aska-SS | Riken | |
| Yamato-SS | Riken | |
| 1273/99 | A gift from Stefan Froehling from NCT Heidelberg | |

| Reagent/resource | Reference or source | Identifier or catalog number |
|---|---|---|
| SK-MES1 | A gift from Stefan Froehling from NCT Heidelberg | |
| MegaX DH10B T1 | Invitrogen/ThermoFisher | |
| **Recombinant DNA** | | |
| pLKO.1 | Addgene | 8454 |
| pMD2 | Addgene | 12259 |
| psPAX2 | Addgene | 12260 |
| **Antibodies** | | |
| SS18::SSX fusion | Cell Signaling Technologies | #70929 |
| SUMO2/3 | Abcam | #ab81371 |
| H2AK119ub | Cell Signaling Technologies | #8240 |
| Vinculin | Abcam | ab130007 |
| b-Actin | Abcam | Ab8226 |
| Anti-rabbit IgG-HRP | ThermoFisher | 31460 |
| Anti-mouse IgG-HRP | ThermoFisher | 31430 |
| **Oligonucleotides and other sequence-based reagents** | | |
| SUMO2_shRNA1 | GCCTGCTTAGAAGTAACATTT | |
| SUMO2_shRNA2 | CCTATTGTGAACGACAGGGAT | |
| SUMO2_shRNA3 | GACTGAGAACAACAATCATAT | |
| SUMO2_shRNA4 | GAGGCAGATCAGATTCCGATT | |
| SUMO2_shRNA5 | CATCCTGACTACTACCGTATA | |
| **Chemicals, enzymes and other reagents** | | |
| Esp3I | NEB | R0734S |
| Polyethylenimine | VWR | 9002-98-6 |
| polybrene | Millipore Sigma | TR-1003 |
| puromycin | Millipore Sigma | P7255 |
| Matrigel | Corning | 356234 |
| DNase I | NEB | M0303L |
| Collagenase II | ThermoFisher | 17101015 |
| NEBNext Ultra II Q5 Master Mix | NEB | |
| DPBS | ThermoFisher | 14190144 |
| DMEM | ThermoFisher | 11965092 |
| Lenti-X Concentrator | Takara Bio | 631232 |
| TrypLE Express Enzyme | ThermoFisher | 12605010 |
| TAK-981 | MedChemExpress | HY-111789 |
| DMSO | Fisher Scientific | BP231-100 |
| Crystal Violet | | |
| L-glutamine | Gibco | 11965092 |
| Antimicotic | Gibco | 15240062 |
| FBS | Gibco | 16140071 |
| Trypsin | Gibco | 25300054 |
| Mammocult medium | StemCell Technologies | 50620 |
| Hydrocortisone | StemCell Technologies | 07925 |

| Reagent/resource | Reference or source | Identifier or catalog number |
|---|---|---|
| Heparin | StemCell Technologies | 07980 |
| Staurosporine | Selleckchem | S1421 |
| Dispase | Gibco | 17105041 |
| Propidium iodide | ThermoFisher | 00-6990-50 |
| Trizol | ThermoFisher | 15596026 |
| Lipofectamine 2000 | ThermoFisher | 11668500 |
| Protoscript II | NEB | E6560S |
| RIPA lysis buffer | ThermoFisher | 89900 |
| protease inhibitor cocktail | ThermoFisher | 78429 |
| LDS Sample buffer | ThermoFisher | J61942.AD |
| 10X reducing agent | ThermoFisher | NP0004 |
| 4–12% Bis-Tris gradient gels | ThermoFisher | NW04120BOX |
| Nitrocellulose membranes | ThermoFisher | IB23001 |
| TBST | ThermoFisher | J77500.K8 |
| SuperSignal West Femto | ThermoFisher | 34094 |
| HPBCD | MedChemExpress | HY-101103 |
| DAB (3, 3 -diaminobenzidine) | Sigma-Aldrich | SK-4105 |
| **Software** | | |
| MAGeCK | Li et al, 2014 | |
| Cutadapt v2.3 | https://cutadapt.readthedocs.io/en/stable/guide.html | |
| Bowtie2 version 2.2.5 | Langmead et al, 2012 | |
| Deeptools alignmentSieve version 3.4.3 | Ramírez et al, 2014 | |
| Picard MarkDuplicates version 2.22.0. | | |
| Macs2 version 2.2.9.1 | Zhang et al, 2008 | |
| Bedtools intersect version 2.29.2 | Quinlan and Hall, 2010 | |
| Homer annotatePeaks.pl | Heinz et al, 2010 | |
| Xenome version 1.0.0 | Conway et al, 2012 | |
| STAR aligner v2.7.0d_0221 | A Dobin et al, 2013 | |
| RSEM v1.3.1 | B Li et al, 2011 | |
| MultiQC v1.8 | P Ewels et al, 2016 | |
| DESeq2 v1.22.2 | M Love et al, 2014 | |
| GSEA app version 4.3.2 | Subramanian et al, 2005 | |
| FastQC v0.11.5 | | |
| **Other** | | |
| Zymo Quick DNA miniprep kit | Zymo | D3024 |
| Illumina HiSeq | Illumina | |

| Reagent/resource | Reference or source | Identifier or catalog number |
|---|---|---|
| CellTiter-Glo Luminescent Cell Viability Assay | Promega | G7570 |
| CellTiter-Glo 3D | Promega | PRG9683 |
| Annexin V-FITC | ThermoFisher | BMS147FI |
| NEBNext Ultra II RNALibrary prep kit for Illumina | NEB | E7770S |
| 2x75bp High Output Cloudbreak Freestyle Kit | Elcesement Biosciences | 860-00034 |
| CUTANA ChIC/ CUT&RUN kit | Epicypher | 14-1048 |

## DepMap data mining and library construction

To identify SS-specific dependencies, we filtered the DepMap CRISPR as well as the RNAi Achilles dataset for genes that were more dependent on growth for synovial sarcoma cell lines when compared to all other cancer cell lines. We then selected the top 200 genes from each dataset which resulted in 348 unique genes in the combined dataset. sgRNA for these genes were designed using CRISPick tool (Doench et al, 2016) from the Broad Institute. sgRNA libraries were synthesized using Array technology (CustomArray, Inc.) containing 3665 guides targeting 348 genes along with 174 guides as non-targeting controls. The guides were amplified by PCR and cloned into pKLO.1 by ligation using the Esp3I (NEB) restriction sites (Wang et al, 2016b, 2016a). Transformations were performed with Invitrogen's MegaX DH10B T1 electro-competent cells using an Eppendorf electroporator 2510 and Bio-Rad Gene Pulser 1 mm cuvettes. A minimum of 30 million successfully transformed cells or 30,000× coverage of the library was obtained.

## Cell culture

HEK-293T and SYO1 cells were cultured in DMEM supplemented with 10% FBS, 1% penicillin–streptomycin and 1% L-glutamine. HS-SY-II and HS-SY-II-Cas9 cells were additionally supplemented with 0.5% Sodium Pyruvate. Aska-SS and Yamato-SS cells were maintained in DMEM supplemented with 20% FBS, 1% penicillin–streptomycin and 1% L-glutamine. The 1273/99 cell line was cultured in DMEM supplemented with F12. All cell lines were authenticated by STR profiling.

## Virus production

Lentivirus was produced in HEK293T cells. Cells from four 80% confluent 10-cm Petri dishes were transfected with 0.9 μg VSV-G envelope expressing plasmid pMD2 and 9 μg psPAX2 packaging vectors and 9 μg of the gRNA library DNA in the presence of 113.4 μL polyethylenimine - PEI (VWR International, 1 mg/mL) per plate. Medium was exchanged after overnight incubation and virus supernatant was collected after 48 and 72 h, passed through a 0.45-μm filter and concentrated by centrifuging at $6000 \times g$ for 2 h at 4 °C. Supernatant was discarded, and pellets were resuspended in 1/1000th volume of PBS and rotated at 4 °C overnight. The concentrated virus was flash frozen in ethanol-dry ice bath and stored at −80 °C.

## In vitro and in vivo CRISPR/Cas9 screens

Screens were performed in duplicates. HS-SY-II Cas9 cells were transduced with the screen library in the presence of 0.8 mg/ml polybrene with an efficiency of 30% or lower to ensure most cells received a single sgRNA. After selection with puromycin (1 μg/mL) for 2–4 days, a cell aliquot containing 5 million cells (~1000× coverage of library) was frozen as the day 0 or input reference sample. The remaining cells were divided into 2 arms for the in vivo and in vitro screens. For the in vivo screen, 2 million cells in 50% Matrigel were transplanted subcutaneously into the flanks of 4 athymic nude mice per replicate. The resultant tumor was monitored, and mice were sacrificed when the tumor volume reached 1 cm³. The tumor was dissociated into single cell suspension (Leelatian et al, 2017) using collagenase II (20 mg/mL) along with Dnase I (10,000 Kunitz/mL) and used for further experiments. For the in vitro screens, at least 5 million cells were maintained throughout the 14-day culture period and collected at the end of the screen. Genomic DNA was extracted from collected cell pellets using a Zymo Quick DNA miniprep kit (#D3024). The sgRNA were PCR amplified by NEBNext Ultra II Q5 Master Mix (NEB #M0544) from the genomic DNA using the indexed PCR primers with next-generation sequencing adapters compatible with Illumina's NEXTERA kit. PCR products were size-selected by gel electrophoresis, quantified by Qubit (ThermoFisher Scientific) and sequenced using HiSeq (Illumina).

## Proliferation assay

The proliferation assay was performed using CellTiter-Glo Luminescent Cell Viability Assay (Promega #G7570) using manufacturer's instructions. Cell numbers were optimized for 384-well plate for each cell line. SUMO2 inhibitor TAK-981 dissolved in DMSO were echo dotted on to a 384-well plate in varying concentrations with the final concentration of DMSO at 0.08% in each well. In total, 25 μl of 50,000 cells/ml were seeded in each well of a 384-well plate. The cells were incubated at 37 °C at 5% $CO_2$ for 48 h then quenched with CellTiter-Glo®, centrifuged at 1000 rpm for 1 min and incubated at RT for 20 min. Luminescence was recorded with a plate reader (BMG FLUOStar). $EC_{50}$ values were calculated by GraphPad Prism software.

## Colony-forming assays

Crystal violet colony-forming assays were conducted by seeding cells at low density in a 6-well plate. After the cells adhered to the plate, they were treated to varying concentrations of TAK-981 for an additional 2–4 days. The wells were then washed, fixed and stained with 0.02% crystal violet solution in methanol. Subsequently, wells were imaged for quantification.

## 2D and 3D cell culture and TAK-981 treatment

To culture cells in 2D and 3D growth formats, SYO1 and ASKA-SS cells were grown in high-glucose DMEM supplemented with 1% L-glutamine (Gibco #11965092) and 1% antibiotic-antimycotic

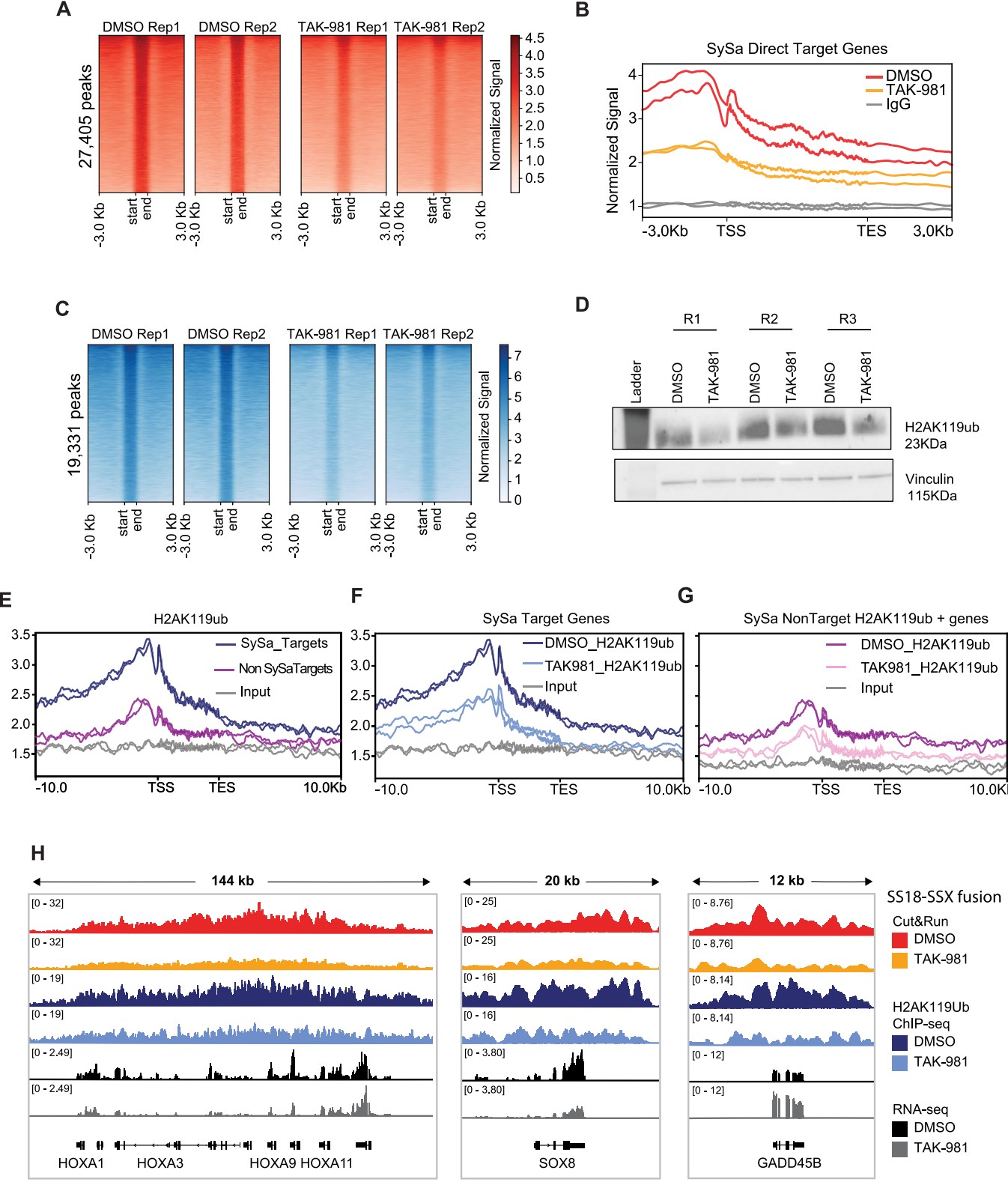

**Figure 6. Treatment with TAK-981 causes the fusion oncoprotein SS18::SSX1 eviction from chromatin.**

(A) CUT&RUN density heatmaps of SS18::SSX occupancy in SYO1 cells treated with DMSO (left) or TAK-981 (right) for 72 h. Across 27,405 peaks are shown. $N = 2$. (B) Meta-analysis plot showing normalized SS18::SSX binding signal ($Y$ axis) at gene bodies from the transcription start site (TSS) to the transcription end site (TES) centered around the TSS $+/- $ 3Kb for SySa direct target genes is shown. DMSO (red), TAK-981 treated (orange), IgG control (grey). (C) ChIP-seq density heatmaps of H2AK119ub occupancy in SYO1 cells treated with DMSO or TAK-981 for 72 h. across 19,331 peaks are shown. (D) Immunoblot of H2AK119ub on SYO1 cells treated with TAK-981 compared to DMSO control are depicted. Vinculin is shown as loading control. $N = 3$ biological replicates. Meta-analysis plot showing normalized H2AK119ub signal at the genomic loci of SySa target genes when compared to the whole genome $+/- $ 10Kb is shown. SySa taget genes (dark blue), rest of the genome (red), IgG control (grey). (E) Meta-analysis plot showing normalized H2AK119ub signal at the genomic loci of SySa target genes when compared to DMSO and TAK-981 treated cells. DMSO (dark blue), TAK-981 treatment (light blue) and IgG control (grey). (F) Meta-analysis plot showing normalized H2AK119ub signal in expression-matched but non-targeting SySa genes. DMSO (dark red), TAK-981 treatment (pink) and IgG control (grey). (G) Integrated genome viewer (IGV) tracks for the SS18::SSX fusion and H2AK119ub in DMSO or TAK-981 treated SYO1 cells along with corresponding RNAseq tracks are shown. (H) For genes in the HOX cluster (left), SOX8 (middle) and GADD45B (right) with tracks for DMSO and TAK-981 treated CUT&RUN on SS18::SSX fusion monoclonal antibody in red and orange, DMSO and TAK-981 treated ChIP-Seq on H2AK119ub in dark and light blue and RNA-seq tracks for DMSO and TAK-981 treated cells in black and grey. Source data are available online for this figure.

(Gibco #15240062) and supplemented with 10% and 20% FBS (Gibco #16140071), respectively. Cells were trypsinized (Gibco #25300054) and counted using Cellometer Auto 2000 (Nexcelom). Mammocult medium (StemCell Technologies #50620) with the addition of 0.5% Hydrocortisone (StemCell Technologies #07925) and 0.2% Heparin (StemCell Technologies #07980) was used as described previously (Al Shihabi et al, 2022).

For the 2D experiments, cells were resuspended in Mammocult at a concentration of 50,000 cells/ml. In total, 100 µl of the solution was dispensed in each well of a 96-well plate. For 3D experiments, cells were resuspended at a concentration of 500,000 cells/ml in a 3:4 mixture solution of Mammocult medium and Matrigel (Corning #354234). The mixture was kept on ice throughout the seeding process. In all, 10 µl of this solution was dispensed around the perimeter of each well's bottom of a 96-well plate to create mini-rings as established previously (Phan et al, 2019; Nguyen and Soragni, 2020; Tebon et al, 2023; Al Shihabi et al, 2022). After a 30-minute incubation at 37 °C to solidify the gel, 100 µl of pre-warmed Mammocult medium to was added to each well using an automated fluid handler (Microlab NIMBUS, Hamilton). In all cases, plates were imaged in brightfield mode every 24 h using a high-content microscope (Celigo, Nexcelom).

Plates were incubated for 2 days before initiating drug treatment. Pre-warmed Mammocult medium containing TAK-981 (MedChemExpress #HY-111789) at six different concentrations diluted in DMSO (Fisher Scientific #BP231-100) was added to the plates after complete removal of media. Each plate included 10 µM staurosporine (Selleckchem #S1421) and 1% DMSO as positive and negative controls respectively.

Treatment was repeated twice or three times after subsequent 24-h incubations. Cell viability was measured after 2 days (post-two total treatments), 3 or 4 days (post-three total treatments) of incubation with TAK-981. Viability was assessed via ATP-release assay (CellTiter-Glo 3D, Promega #PRG9683) after PBS washes (Gibco #14190144) and incubation with 50 µl of dispase (Gibco #17105041) at 37 °C for 25 min. Plates were incubated in the dark at room temperature for 25 min upon addition of the CellTiter Glo 3D reagent. Luminescence was measured using a SpectraMax iD3 plate reader (Molecular Devices). The viability of each well was normalized to the vehicle control wells.

### Apoptosis and cell cycle assays

Apoptosis was quantified by flow cytometry using Annexin V-FITC kit from BD Biosciences. In total, $3 \times 10^5$ SYO1 and HS-

SY-II cells were seeded in a six-well plate and allowed to attach for 24 h. TAK-981 was added in varying concentrations and incubated for 48 h. After incubation, the cells were trypsinized, washed in warm PBS, and resuspended in Annexin V binding buffer. Annexin V-FITC was added and incubated at room temperature for 10 min. The samples were then analyzed by flow cytometry using Fortessa (BD Bioscience, USA) along with FlowJo analysis software.

### Cell cycle analysis

Cell cycle analysis was done by staining the cells with propidium iodide (PI). As previously stated, $3 \times 10^5$ SYO1 and HS-SY-II cells were seeded in a six-well plate and allowed to attach for 24 h. They were then exposed to varying concentrations of TAK-981 for 48 h. Cells were trypsinized, washed with PBS, and fixed with ethanol. Cells were washed and stained with PI. The samples were then analyzed by flow cytometry using Fortessa (BD Bioscience, USA) along with FlowJo analysis software.

### RNA sequencing

HS-SY-II and SYO1 cells were treated with either TAK-981 at concentrations of 25 nM and 100 nM respectively for the treatment arm or DMSO for the control arm for 48 h. Cells were pelleted and RNA was extracted using Trizol (Thermo, Cat. No. 15596026) with concentration determined by Qubit (Thermo Scientific). Libraries were prepared with the NEBNext Ultra II RNALibrary prep kit for Illumina (NEB, Cat. No. E7770S).

### Small hairpin RNA (shRNA) transfection and transduction

Small hairpin RNA (shRNA) for SUMO2 were cloned into the all-in-one-Tet vector and packaged into lentivirus using pMD and pPax2 as described above. 300,000 HS-SY-II and SYO1 cells were seeded into six-well culture plates overnight. Lipofectamine 2000 reagent (Invitrogen, Waltham, MA, USA) was used to perform the transfections, as described in the manufacturer's instructions. At 48 h after transfection, media was changed and puromycin selected for 2 days. After selection, cells were subjected to a previously determined amount of doxycycline (4.5 µg/ml) for 48 h. Cells were harvested for qPCR quantification and western blot analysis.

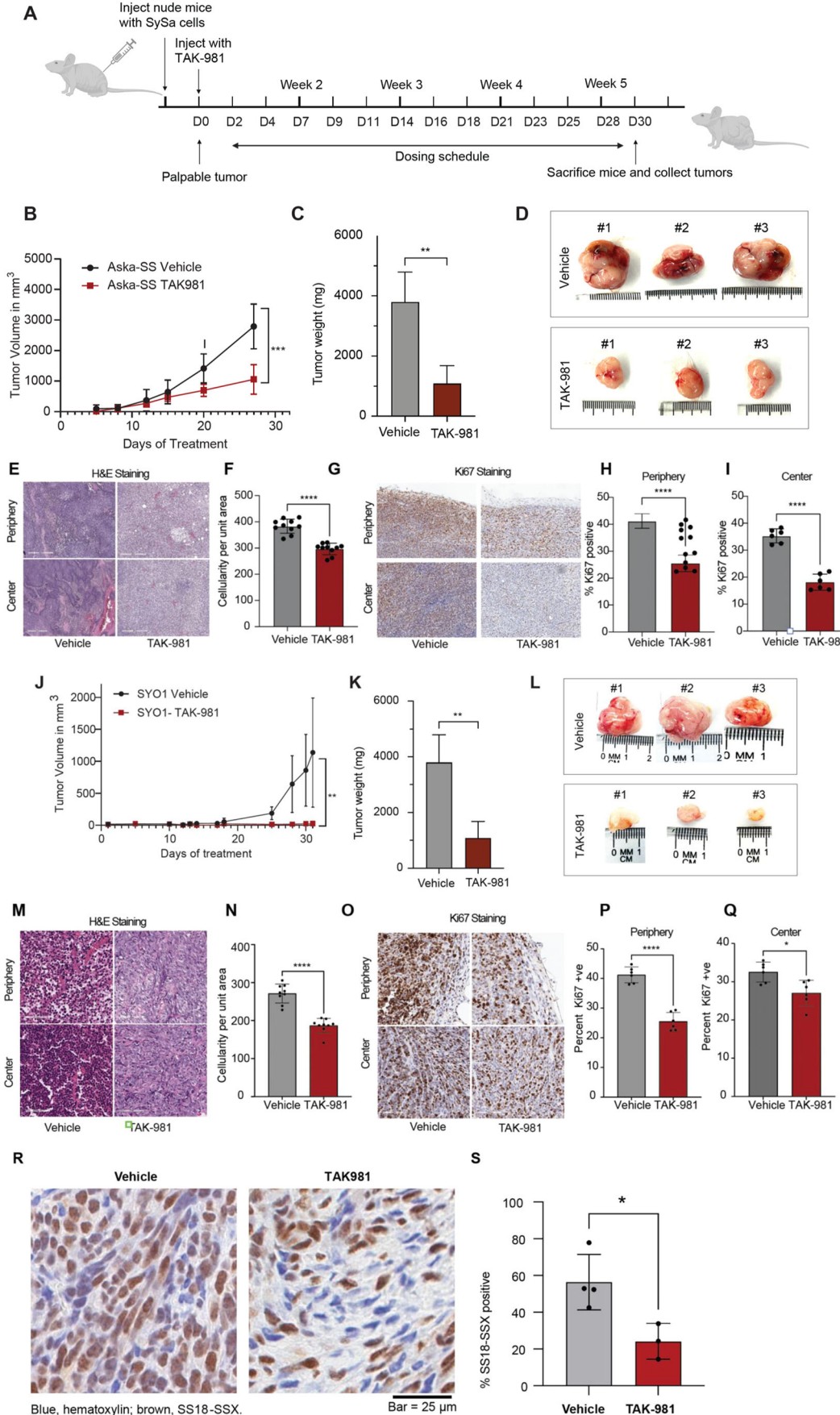

**Figure 7. Tumor size reduction is seen in mice treated with TAK-981.**

(A) Schematic showing the in vivo experiment with TAK-981 treatment in Aska-SS or SYO1 injected nude mice. The cartoon depicts time of cell injection, duration and frequency of treatment, and time of final tumor harvesting. (B) Average tumor volumes of mice injected with Aska-SS are shown for the duration of the experiment. $N = 3$ biological replicates per condition, $P$ values were calculated using two-way ANOVA with $P = 0.0002$. Error bars represent mean ± SD. (C) Tumor weights for DMSO or TAK-981 treated Aska-SS mice are shown. $N = 3$ biological replicates per condition, $P$ values were calculated using a two-tailed Student's $T$ test **$P < 0.01$ ($P = 0.0053$). Error bars represent mean ± SD. (D) Representative images of extracted Aska-SS tumors. (E) Immunohistochemical staining of tumors with hematoxylin-eosin (H & E) staining reflecting tumor areas from vehicle treated and TAK-981 treated Aska-SS-injected mice. Scale bar = 300 μm. (F) Quantification of tumor cellularity in vehicle-treated and TAK-981-treated Aska-SS injected groups using ImageJ. $N = 10$ fields per sample, $P$ values were calculated using a two-tailed Students $T$ test, ****$P < 0.0001$ ($P = 0.0000003160$). Error bars represent mean ± SD. (G) Representative IHC images showing Ki67 staining in the vehicle and TAK-981 treated Aska-SS tumor sections from the periphery (top) and center (bottom) of tumors. Staining indicates Ki67-positive cells—a marker of proliferation. Scale bar = 300 μm. (H, I) Quantification of images in (G) using ImageJ. $N = 6$ fields per sample. $P$ values were calculated using a two-tailed students T-test, ****$P < 0.0001$ ($P$ value for H = 0.00000256, and $P$ value for I = 9.46E-07). Error bars represent mean ± SD. (J–Q) Similar results are shown for SYO1 injected tumors, (J) shows tumor volumes over time $N = 7$ biological replicates, $P$ values were calculated using two-way ANOVA with $P = 0.0012$. Error bars represent mean ± SD. (K) shows tumor weight quantification with $N = 5$ biological replicates, $P$ values were calculated using a two-tailed Student's $T$ test, **$P < 0.01$ ($P = 0.0422$). Error bars represent mean ± SD. (L) representative tumor images, (M) displays H&E staining in Vehicle-treated compared to TAK-981-treated SYO-1 injected mouse tumors. Scale bar = 60 μm. (N) shows the quantification of cellularity per unit area, using ImageJ. $N = 10$ fields per sample, $P$ values were calculated using a two-tailed Students $T$ test, ****$P < 0.0001$ ($P = 0.0000004856$). Error bars represent mean ± SD. (O–Q) shows pictorial (scale bar = 60 μm) and quantitative depiction of Ki67 staining in the periphery or center of the tumors in SYO1 injected mice with TAK-981. (P) Quantification of the tumor periphery using ImageJ. $N = 10$ fields per sample, $P$ values were calculated using a two-tailed Students $T$ test, ****$P < 0.0001$ ($P = 0.000003$). Error bars represent mean ± SD. (Q) Quantification of the center of the tumor using ImageJ. $N = 10$ fields per sample, $P$ values were calculated using a two-tailed Students $T$ test, *$P < 0.01$ ($P = 0.0101$). Error bars represent mean ± SD. (R) Representative IHC images showing SS18::SSX fusion-specific antibody stained along with hematoxylin in the vehicle and TAK-981 treated SYO1 tumor sections are shown. (S) Quantification of cells stained positively stained for SS18::SSX fusion in vehicle-treated and TAK-981-treated SYO1 injected groups. $N = 3$ biological replicates, $P$ values were calculated using a two-tailed Students $T$ test, *$P < 0.05$ ($P = 0.0241$). Error bars represent mean ± SD. Source data are available online for this figure.

## Quantitative real-time reverse transcription PCR (qRT-PCR)

TRIzol Reagent (ThermoFisher) was used to extract total RNA from SYO1 and HS-SY-II cell pellets and 1st strand was synthesized using Protoscript II (NEB) with polyA selection. qPCR was performed using TaqMan Gene Expression Master Mix and FAM probes for SUMO2, HPRT, GAPDH from ThermoFisher.

## Western blot analysis

Whole cell lysates from synovial sarcoma cancer cell lines treated with varying concentrations of TAK-981 or with 0.1% DMSO were prepared on ice with RIPA lysis buffer (Thermo #89900) supplemented with protease inhibitor cocktail (Thermo #78429). Lysates along with LDS Sample buffer (Thermo #J61942.AD) were heated at 65 °C for 10 min. Proteins were separated on precast 4–12% Bis-Tris gradient gels (Thermo #NW04120BOX). Separated proteins were subsequently transferred to nitrocellulose membranes (Thermo #IB23001) using the iBLOT2 system. Membranes were blocked with PBS containing 5% milk powder and 0.05% Tween-20 for 1 h. Protein samples were incubated with primary antibodies against SS18::SSX fusion at 1:1000 dilution (rabbit monoclonal from Cell Signaling Technologies #70929), SUMO2/3 at 1:500 dilution (mouse monoclonal 8A2 from Abcam #ab81371) and H2AK119ub at 1:1000 dilution (rabbit monoclonal from Cell Signaling Technologies #8240). Mouse anti-Vinculin monoclonal antibody (Abcam #ab130007) was used as a loading control. (Goat anti-rabbit IgG-HRP and goat anti-mouse IgG-HRP were used as secondary antibodies at 1:3000 dilution in 5% milk. Signal was detected using SuperSignal West Femto (Thermo # 34094) and captured using the Bio-Rad ChemiDoc system (Cat. No. 1708370).

## Mouse experiments

In this study, 8- to 10-week-old male Nu/J mice were acquired from Jackson Laboratory. The animals were housed in individually ventilated cages under specified pathogen-free conditions in the animal facilities of our institute. All animal experiments were conducted in accordance with protocols approved by the institutional animal care and use committee at the Sanford Burnham Prebys Medical Discovery Institute.

Two million Aska-SS or SYO1 cells were injected subcutaneously into the right flank of each mouse in a mixture of 100 μL PBS and 50% Matrigel (Corning #356234). Once tumors were established, typically within 3–8 days post-implantation, drug treatment was initiated. Mice received 0.25 mL of 25 mg/kg TAK-981 in 20% HPBCD or a vehicle control via intraperitoneal injection three times per week. Tumor growth was monitored two to three times per week using a vernier caliper and imaging until ethical endpoints necessitated euthanasia due to tumor size or ulceration. Tumor volume was calculated using the formula: volume (V) = $W^2 \times L/2$, where W is the width and L is the length of the tumor.

## Immunohistochemistry

Tumors excised from nude mice were formalin-fixed and paraffin-embedded. Sections of 5 μm were cut and mounted on slides (Medline Cat.No. MLABSLIDE1WC). After deparaffinization, antigen retrieval was carried out in PBS (pH 6) using a pressure cooker for 10–15 min. Tissue sections were blocked with 10% donkey serum for an hour and incubated with the primary antibody at 4 C overnight. After multiple PBS washes, the sections were incubated with the secondary antibody for 45 min at room temperature. Visualization was performed using HRP substrate DAB (3, 3 -diaminobenzidine) (Cat. No. SK-4105). Sections were counterstained with hematoxylin. On the scanned slides, the percentage of nuclear SS18::SSX positive cells was quantified using QuPath's (Bankhead et al, 2017) positive cell detection function.

## ChIP-seq

SYO1 cells were treated with DMSO (control) or 1 μM TAK-981 for 72 h. ChIP-seq was performed to assess changes in histone 2 A

ubiquitination at lysine 119 (H2AK119ub) as described earlier (Barbosa et al, 2024). SYO1 cells plated and treated in 10-cm tissue culture treated plates in triplicates for each group (DMSO and TAK-981) were trypsinized and counted for fixing after 72 h treatment. 1 million cells from each plate were fixed using 1% formaldehyde for 10 min at room temperature. Fixed cells were sheared using Bioruptor (Diagenode, NJ) in 15 cycles, each with 30 s. on and 30 s. off settings at 4 °C. Chromatin was immunoprecipitated using antibody for H2AK119ub (Cell Signaling Technologies # 8240). DNA was purified after reverse crosslinking. Immunoprecipitated chromatin was subjected to library prep using NEBNext Ultra II DNA library prep kit for Illumina (E7645S and E7600S) as per the manufacturer's protocol. Library prepped DNA then sequenced on AVITI platform (Element Biosciences) with the 2x75bp High Output Cloudbreak Freestyle Kit.

## CUT&RUN

Changes in genome wide binding of the SS18::SSX2 fusion after TAK-981 treatment were studied in SYO1 cell line using CUT&RUN assay. SYO1 cells were treated in duplicates with DMSO or 1 μM TAK-981 for 72 h. At 72 h, cells were trypsinized, washed with PBS and counted for the assay. In total, 300,000 cells per antibody were then bound on activated ConA magnetic beads and CUT&RUN was performed using the CUTANA ChIC/CUT&RUN kit (Epicypher, NC # 14-1048) as per the manufacturer's protocol. Permeabilized cells were incubated with anti-Rabbit IgG (Epicypher, #13-0042) or SS18::SSX (Cell Signaling technologies # 72364) overnight at 4 °C. K-MetStat panel (provided in the kit, #19-1002) was added to IgG control samples. E.coli Spike-In DNA, also provided in the kit (#18-1401) was added to each sample as mentioned in the protocol. Purified DNA then subjected to library prep using NEBNext Ultra II DNA library prep kit for Illumina (E7645S and E7600S) and sequenced on Element Biosciences AVITI platform with the 2x75bp High Output Cloudbreak Freestyle Kit.

## Data analysis

### Pooled CRISPR screen
MAGeCK (Model-based Analysis of Genome-wide CRISPR-Cas9 Knockout) (Li et al, 2014) pipeline was used for mapping reads (paired end fastqs) to sgRNA custom library (Appendix).

### CUT&RUN
Paired-end reads were trimmed using Cutadapt version 2.3 (Martin, 2011) with parameters "-j 12 -m 20 -O 5 -q 15 -a AGATCGGAAGAGCACACGTCTGAACTCCAGTCAC -A AGATCGGAAGAGCGTCGTGTAGGGAAAGAGTGT". Trimmed reads were aligned against E. coli genomic sequence (GCF_000005845.2_ASM584v2_genomic) using Bowtie2 version 2.2.5 (Langmead and Salzberg, 2012) with parameters "--local" to quantify spike-in amount. Unmapped reads were subsequently aligned against hg38 chrM to remove mitochondrial reads using Bowtie2 with parameters "--local -X 2000". Remaining unaligned reads were mapped against human genome version hg38 (without chrM) using Bowtie2 with parameters "--very-sensitive --no-discordant -X 2000". Multimapping and improperly paired reads were removed using Deeptools alignmentSieve version 3.4.3 (Ramírez et al, 2014) with parameters "--minMappingQuality 30 --samFlagInclude 2". Duplicate reads were removed using Picard MarkDuplicates

version 2.22.0. Peak calling was performed in DMSO treated samples using Macs2 version 2.2.9.1 (Zhang et al, 2008) with parameters "--nomodel --shift -75 --extsize 150 --keep-dup all -q 0.01 --broad --broad-cutoff 0.1 --gsize 2700000000.0 --format BAMPE". Overlapping peaks in DMSO replicates were determined using Bedtools intersect version 2.29.2 (Quinlan and Hall, 2010). Overlapping peaks were merged using Bedtools merge and parameter "-d 500". Peaks overlapping Encode blacklist regions were removed. Peaks were annotated and peak tags were counted in each sample using Homer annotatePeaks.pl (Heinz et al, 2010) with parameters "hg38 -raw". Bigwig signal files were generated using Deeptools bamCoverage with parameters "--binSize 20 --smoothLength 500 -p 12 --normalizeUsing RPGC --extendReads --ignoreForNormalization chrX --effectiveGenomeSize 2913022398 – scaleFactor [spike-in scale factor]".

### ChIP-seq
Mouse spike-in reads were classified and separated from SYO1 ChIP-seq reads using Xenome version 1.0.0 (Conway et al, 2012). First mate (read1) of each sample were trimmed using Cutadapt version 2.3 (Martin, 2011) with parameters "-j 12 -m 30 -O 5 -q 15 -a AGATCGGAAGAGCACACGTCTGAACTCCAGTCAC". Trimmed reads were aligned against hg38 chrM to remove mitochondrial reads using Bowtie2 (Langmead and Salzberg, 2012) with parameters "--local". Remaining unaligned reads were mapped against human genome version hg38 (without chrM) using Bowtie2 with parameters "--local". Multimapping reads were removed using Deeptools alignment Sieve version 3.4.3 (Ramírez et al, 2014) with parameters "--minMappingQuality 30". Duplicate reads were removed using Picard MarkDuplicates version 2.22.0. Peak calling was performed using Macs2 version 2.2.9.1 (Zhang et al, 2008) with parameters "--keep-dup all -q 0.01 --broad --broad-cutoff 0.1 --gsize 2700000000.0 --format BAM". Overlapping peaks in DMSO replicates were determined using Bedtools intersect version 2.29.2 (Quinlan and Hall, 2010). Overlapping peaks were merged using Bedtools merge. Peaks overlapping Encode blacklist regions were removed. Peaks were annotated and peak tags were counted in each sample using Homer annotatePeaks.pl (Heinz et al, 2010) with parameters "hg38 -raw". Bigwig signal files were generated using Deeptools bamCoverage with parameters "--binSize 20 --smoothLength 500 -p 12 --normalizeUsing RPGC --ignoreForNormalization chrX --effectiveGenomeSize 2913022398.

### RNA-seq data analysis
Raw reads were preprocessed by trimming Illumina Truseq adapters, polyA, and polyT sequences using cutadapt v2.313 with parameters "cutadapt -j 4 -m 20 --interleaved -a AGATCGGAAGAGCACACGTCTGAACTCCAGTCAC -A AGATCGGAAGAGCGTCGTGTAGGGAAAGAGTGT Fastq1 Fastq2 | cutadapt --interleaved -j 4 -m 20 -a "A{100}" -A "A{100}" - | cutadapt -j 4 -m 20 -a "T{100}" -A "T{100}" -". Trimmed reads were subsequently aligned to human genome version hg38 using STAR aligner v2.7.0d_0221 14 with parameters according to ENCODE long RNA-seq pipeline (https://github.com/ENCODE-DCC/long-rna-seq-pipeline). Gene expression levels were quantified using RSEM v1.3.1 15. Ensembl v84 gene annotations were used for the alignment and quantification steps. RNA-seq sequence, alignment, and quantification qualities were assessed using FastQC v0.11.5 (https://www.bioinformatics.babraham.ac.uk/projects/fastqc/) and MultiQC

v1.8 16. Lowly expressed genes were filtered out by retaining genes with estimated counts (from RSEM) ≥ number of samples times 5. Filtered estimated read counts from RSEM were used for differential expression comparisons using the Wald test implemented in the R Bioconductor package DESeq2 v1.22.2 based on generalized linear model and negative binomial distribution 17 (Love et al, 2014). Genes with Benjamini–Hochberg corrected $P$ value < 0.05 and fold change ≥2.0 or ≤2.0 were selected as differentially expressed genes. Gene set enrichment analysis (GSEA) was performed using GSEA app version 4.3.2 (Subramanian et al, 2005).

### Statistical analysis

All statistical analysis was performed using GraphPad Prism 9.0. Data were presented as the mean ± SEM. Statistical significance between two groups was determined using Students $t$ test. Significance over multiple time points among groups was computed using two-way ANOVA. Dose response curves were fit using four parameter logistic equation. A statistical threshold of $P < 0.05$ was used with $^*P < 0.05$; $^{**}P < 0.01$; $^{***}P < 0.001$; $^{****}P < 0.0001$; $P$ = ns, not significant.

### Graphics

Some of the figures were created with BioRender.com.

## Data availability

Sequencing data for RNA-seq, CUT&RUN, ChIP-seq, and high-throughput CRISPR screens, are deposited in the NCBI GEO under accession number: GSE276074. Code for data analysis is available in the Github page: https://github.com/PBioinfo/Synovial_Sarcoma_Paper_code.git.

The source data of this paper are collected in the following database record: biostudies:S-SCDT-10_1038-S44318-025-00526-w.

## Peer review information

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

## Acknowledgements

We would like to thank Adriana Charbono and Buddy Charbono for their invaluable assistance with mouse studies, Dr. Chih-Cheng Yang and Chun-

Teng Huang from the Sanford Burnham Prebys Medical Discovery Institute (SBP) functional genomics core, Yoav Altman from the SBP Flow Cytometry Core, and Drs. Rebecca Porritt and Kang Liu from the Genomics Core, Guillermina Garcia from the Histology core for their excellent support. We would like to acknowledge the help of Dr. Derron Herr and Anis Shahnaee from the Jerold Chun Lab at SBP for their assistance with microscopy. This work was supported by National Institutes of Health (NIH) National Cancer Institute grants CA262746 and P30 CA030199. We would also like to acknowledge the support of the Animal Facility, SBP Flow Cytometry Core, Genomics core, Histology core, Functional Genomics core as well as the Bioinformatics core supported by the NCI Cancer Center Support Grant P30 CA030199.

## Author contributions

**Rema Iyer**: Conceptualization; Data curation; Formal analysis; Validation; Investigation; Visualization; Methodology; Writing—original draft; Project administration; Writing—review and editing. **Anagha Deshpande**: Conceptualization; Data curation; Formal analysis. **Aditi Pedgaonkar**: Conceptualization; Data curation; Formal analysis. **Pramod Akula Bala**: Data curation; Software; Formal analysis; Methodology. **Taehee Kim**: Conceptualization; Data curation; Formal analysis. **Gerard L Brien**: Resources. **Darren Finlay**: Data curation; Formal analysis; Methodology. **Kristiina Vuori**: Resources; Funding acquisition. **Alice Soragni**: Conceptualization; Resources; Funding acquisition. **Hiromi I Wetterstein**: Conceptualization; Formal analysis; Methodology. **Rabi Murad**: Data curation; Software; Formal analysis; Methodology. **Aniruddha J Deshpande**: Conceptualization; Resources; Supervision; Funding acquisition; Investigation; Visualization; Methodology; Writing—original draft; Project administration.

Source data underlying figure panels in this paper may have individual authorship assigned. Where available, figure panel/source data authorship is listed in the following database record: biostudies:S-SCDT-10_1038-S44318-025-00526-w.

## Disclosure and competing interests statement
The authors declare no competing interests.

