## [Peer Review File · The EMBO Journal]

Targeting SUMO2 Reverses Aberrant Epigenetic Rewiring Driven by SS18::SSX Fusion Oncoproteins and Impairs Sarcomagenesis

Rema Iyer, Anagha Deshpande, Aditi Pedgaonkar, Pramod Bala, Taehee Kim, Gerard Brien, Darren Finlay, Kristiina Vuori, Alice Soragni, Hiromi Wetterstein, Rabi Murad, and Aniruddha Deshpande

Corresponding author: Aniruddha Deshpande (adeshpande@sbsdsc.discovery.org)

Review Timeline:

Transferred from Review Commons:	11th Mar 25
Editorial Decision:	30th Apr 25
Editor's Correspondence:	8th May 25
Revision Received:	19th May 25
Accepted:	23rd Jul 25

Editor: Daniel Klimmeck

Transaction Report:

This manuscript was transferred to The EMBO Journal following peer review at Review Commons.

Review #1**1. Evidence, reproducibility and clarity:****Evidence, reproducibility and clarity (Required)******Summary****

The manuscript by Iyer and colleagues studies the exceptional dependency of synovial sarcoma (SS) cells on SUMO2. This dependency was initially identified by mining the DepMap database for selective genetic dependencies of SS cell lines. Among the combined top 200 SS-selective genes from the DEMETER and CHRONOS datasets, several members of the sumoylation machinery were identified, including SUMO2.

These findings were then first validated through two CRISPR screens in an SS cell line using sgRNAs against all these DepMap hits. The screens were performed both in vitro and in vivo in mice and 19 hits were found to be common to both screens. Of these 19 genes, two were also SS18-SSX targets, including SUMO2. Based on these findings, the effect of the sumoylation inhibitor TAK981 was tested on several SS cell lines. The drug affected cell viability with an EC50 in the nanomolar range in both 2D and 3D culture systems, while in non-SS cell lines it was considerably less potent.

RNAseq was then performed to evaluate the effect of TAK981 on the transcriptome in two cell lines. A total of 1100 genes were found to be affected, including many genes involved in cell cycle regulation and several SS18-SSX target genes. Interestingly, a doxycycline resistance signature was reversed by the treatment. A closer look at the SS18-SSX target genes revealed that TAK981 treatment led to a downregulation of SS18-SSX activated genes and to a reactivation of SS18-SSX repressed genes. To test the hypothesis that SUMO2 regulates the fusion protein itself, the effect of shRNA-mediated silencing of SUMO2 and treatment with TAK981 on its expression was assessed. Both treatments indeed led to a downregulation of SS18-SSX, suggesting that its expression is indeed under the control of SUMO2.

CUT&RUN analysis before and after treatment with TAK981 confirmed the reduction of SS18-SSX at genomic binding sites and also revealed a global reduction of H2AK119ub. Finally, the effect of TAK981 on the growth of cell line-derived xenograft tumors was tested. TAK981 slowed the growth of Aska-SS tumors, while treatment of SYO1 tumors completely prevented its growth.

****Major Comments****

Figure 1

A-B It appears that many genes are unique to one data set or the other. The overlap between the two data sets should be presented in a table or a Venn diagram.

Figure 2

B-C The x-axis of these plots is labelled 'genes' and goes up to 500. However, the sgRNA library was directed against 348 genes, while the rest are non-targeting controls. I suggest that the axis be labelled differently, e.g. sgRNA.

It is not clear why the authors restricted their selection to downstream targets of the fusion protein. This should be explained in the text.

Figure 3

B-D The extent of apoptosis and cell cycle arrest in case of SYO-1 cells is relatively small. The physiological effect behind the decrease in viability (Fig.3A) is therefore not very obvious in this case. Furthermore, it is stated in the text that TAK981 "cell cycle analysis using propidium iodide indicated an S-phase arrest". However, in the treated cells there are less S-phase cells detected. Taken together, it is not clear whether the effect of their drug in this case is mainly cytostatic or cytotoxic. I suggest that a cell death assay be performed, using flow cytometry to detect both apoptotic and dead cells, to determine the effect on cell viability.

E-F These figures show essentially the same data. One of the figures could also be removed. Why are ASKA-SS cells used here and not HS-SY-II cells?

G The cells apparently do not form colonies, similar to other sarcomas. The visible clouds of cells probably originate from uneven seeding. In my opinion, this cannot be called a colony forming assay. I suggest that this panel be removed from the figure.

Figure 4

D The finding that the doxorubicin resistance signature is reversed by TAK981 treatment is interesting. The authors could check the combinatorial effects of TAK981 and doxorubicin experimentally.

Figure 5

G The Western blot contains vertical lines between the two lanes with shRNA-1 treated sample and the rest of the samples, indicating that these samples were run on a different gel and the blot was spliced afterwards. Although the samples with the same shRNA treatment (plus/minus Dox) were obviously run on the same blot, it is necessary to indicate the different blots. This can be done by showing the blots separately next to each other.
G-I Please indicate which band(s) represent SS18-SSX. Does any of the bands correspond to the wild-type protein?

Figure 6

F-G Tracks showing the difference between TAK981 and DMSO would show the effect of the treatment more clearly.
It should be evaluated whether the reduction of H2AK119ub is specific for SS18-SSX binding sites.

Figure 7

B Some tumors in the control group are very large at the experimental endpoint (in the range of 4000 mm³). Is this consistent with their animal license?

****Minor comments****

Figure 3

F Concentration labelling is missing.

Figure 4

Text line 199-203: "Upregulated" and "downregulated" genes are mixed up.

Figure 5

D Is the labelling "direct targets" correct here or are these also indirect targets as in E?

Figure 6

In the text, the immunoblot is described as C, but the figure is actually 6E.

Figure 7

B The x-axis labelling is missing and the y-axis labelling should be mm³.

J The tumors in this experiment were very small at the start of treatment. So it is more of an engraftment assay rather than a tumor growth assay.

2. Significance:

Significance (Required)

General assessment: Many fusion protein-driven sarcomas, including SS, have a poor prognosis and effective new therapies are urgently needed. However, the driver fusion proteins composed of TFs or epigenetic regulators (such as SS18-SSX) are difficult drug targets. Therefore, upstream regulators or downstream effectors that are easier to inhibit are of great therapeutic interest. The identification of SUMO2 as a regulator of SS18-SSX expression in SS cells, which may be the reason for the selective dependence of SS cells on this gene, represents an important advance into this direction.

Strength and limitations: Overall, the study is well conducted, the methodology is mostly appropriate, and the conclusions are reasonable and supported by evidence. An exception however are the physiological experiments, which are rather superficial and do not provide a clear picture of the physiological effects of TAK981. An important ethical criticism concerns the size of the xenograft tumors, which were grown here to 4000 mm³, exceeding the size limit normally considered in ethical guidelines. From a scientific point of view, my overall assessment of this paper is positive, but I think the above mentioned comments need to be addressed.

My own expertise is in the field of translational cancer research with a special focus on sarcoma, the mechanisms of action of oncogenic fusion proteins, generation of new tumor models and identification of novel therapeutics.

3. How much time do you estimate the authors will need to complete the suggested revisions:

Estimated time to Complete Revisions (Required)

(Decision Recommendation)

Between 1 and 3 months

4. Review Commons values the work of reviewers and encourages them to get credit for their work. Select 'Yes' below to register your reviewing activity at Web of Science Reviewer Recognition Service (formerly Publons); note that the content of your review will not be visible on Web of Science.

No

Review #2

1. Evidence, reproducibility and clarity:

Evidence, reproducibility and clarity (Required)

The authors seek to evaluate genetic vulnerabilities in synovial sarcoma in an effort to identify possible new targets for therapies. Synovial sarcoma is a soft-tissue sarcoma that occurs most frequently in children and young adults. Treatment options for metastatic and recurrent diseases are limited, making the work by Iyer and colleagues timely and important. Iyer utilizes several datasets of genomic screens to identify a set of genes and gene pathways enriched in synovial sarcoma compared to other data-sets. With CRISPR screens of these identified genes, they identify SUMO2 as a top candidate hit. They explore the impact of TAK-981, a SUMO2 inhibitor, on synovial sarcoma cell line growth and on overall gene transcription, demonstrating that it specifically modifies the SS18-SSX transcriptomic program. Finally, they explore the role of TAK-981 in vivo.

Overall, Iyer et al demonstrate an intriguing new clinical target for synovial sarcoma therapy. To strengthen and support their analysis, we recommend the following for consideration:

- Throughout the manuscript, the authors use TAK-981 to demonstrate the impact of SUMO2 inhibition. To ensure that the effects on cell growth inhibition and cell cycle arrest are on target, it would be helpful for readers to see the experiments also completed with modulation of SUMO2 through knockdown or knockout.
- The authors utilize two synovial sarcoma cell lines implanted subcutaneously to demonstrate the impact of TAK-981 in vivo. They demonstrate an impact on tumor growth and overall proliferation. However, it would be helpful to the authors to know whether TAK-981 is acting on target in the in vitro studies.
- In Figure 3 Panels B and C, there is no notation of what TAK-981 is used for the data

provided. It is difficult to assess the results without understanding which concentration is used. Moreover, the percent cells staining with Annexin V is quite low in 3B - interpretation of this low percentage would be improved with an understanding of the dose utilized

- For interpretation by the reader, it would be helpful to provide a representative scatter plot or histogram from the flow cytometry in Figure 3 panels B-D

- In Figures 5H-I and 6E, there are no immunoblots provided within the figure.

- In Figure 3F, it is difficult to assess the data due to the compressed nature of the Y axis.

Additionally, the X axis does not provide units

- In Figure 3G, the photos provided of the clonogenic assay are blurry and difficult to assess. Quantification of the experiment is not provided. Additionally, the large cluster of cells at the center of the plate likely limits the ability to identify individual colonies, as is necessary for a clonogenic assay.

- In Figure 5 panel G, the loading control is hard to appreciate for the shRNA4 +dox sample - it is unclear if it is underloaded or if it is the image quality.

- In Fig 1A-B, the authors utilize a word cloud to demonstrate the possible genes identified through their analysis. It would be helpful for authors to understand the difference in p value that is represented by the different font sizes

- In line 299, the second 'in' does not need to be italicized

2. Significance:

Significance (Required)

Synovial sarcomas are lethal soft tissue sarcomas of children and young adults with very limited treatment options, especially for relapsed and metastatic disease. This paper is timely and important given that TAK-981 is already in clinical trials. The mechanistic insights are useful as well and in line with existing evidence of SS being primarily an epigenetically driven cancer. Overall, I am supportive of publication with some revisions.

3. How much time do you estimate the authors will need to complete the suggested revisions:

Estimated time to Complete Revisions (Required)

(Decision Recommendation)

Between 1 and 3 months

4. Review Commons values the work of reviewers and encourages them to get credit for their work. Select 'Yes' below to register your reviewing activity at Web of Science Reviewer Recognition Service (formerly Publons); note that the content of your review will not be visible on Web of Science.

No

Review #3

1. Evidence, reproducibility and clarity:

Evidence, reproducibility and clarity (Required)

Authors use both a DEPmap analysis, their own screen and literature precedents to hone in on SUMO2 as a targetable dependency in synovial sarcoma, a disease with clinical need for new systemic treatments. A new SUMO targeting drug (TAK-981) is then shown to be effective against multiple synovial sarcoma cell lines (and cell line xenografts), in terms of growth/apoptosis assays, reversing disease-specific oncogenic transcription patterns, and decreasing levels of the key synovial sarcoma oncoprotein SS18::SSX and its H2AK119Ub partner in terms of total protein and chromatin occupancy (including at target genes).

****Major comments:****

The supplemental tables S1, S2, S3 and S4 were not included in the materials.

Related re: Fig 1: The Analysis of functional genomic screens does not provide details on what cell lines were included -- in particular which and how many synovial sarcoma cell lines. The supplementary tables which may have had this information were not included with the submitted material nor can they be found on the bioRxiv posted version, but even if provided in supplemental this information should be brought into the main text of methods and/or the first part of the results. It is important that screening results for "synovial sarcoma" are not based on data from the notorious (in the synovial sarcoma field) SW982 cell line. SW982 carries a BRAF mutation characteristic of melanoma and lacks the SS18::SS18 oncogene, and so despite its persistent ATCC annotation as derived in the 1970s from a "synovial sarcoma", this cell line is definitely not a synovial sarcoma. If it has been included, then the analysis has to be redone excluding it and Figure 1 has to be re-rendered.

Re: Fig 3: while the cell lines are described as STR authenticated, it is important that the four synovial cell lines be specifically confirmed to express SS18::SSX (and conversely that the control cell lines do not express this). For these experiments, the number of cell lines and the use of two non SySa cell lines is good. However, in terms of controls for the drug

testing, while the DMSO negative control is important, for context -- in a manuscript claiming to be identifying a new treatment for synovial sarcoma -- there should also be a positive control drug treatment arm - i.e. a comparison of at least some of the assay results shown in this figure to results obtained in the same cell line(s) with a drug currently used for SySa treatment ... like doxorubicin (or trabectedin, or pazopanib). Indeed, based on the penultimate sentence in the discussion, a combo arm of TAK + doxorubicin might also be of interest.

Re: Fig 5. This is a great way to address if TAK-981 is affecting SS18::SSX induced biology. My concern is that perhaps the best supporting data has been cherry-picked and selectively presented. Specifically, the full list of Jerby-Arnon "core oncogenic program" genes should be shown so the reader can assess not only the ones where the expression is reversed, but also for how many of those genes there is actually no effect. The key finding (Fig 5G) that SUMO2 knockdown leads to a decreased level of the SS18::SSX oncoprotein itself is only shown for HS-SY-II cells; this particular finding is especially important to assess in multiple cell lines (as is done for many other less-critical figure panels in this manuscript).

Some of the text in the results section "SUMO2 inhibition diminishes SS18-SSX chromatin occupancy..." describing Figure 6 does not match what is actually presented in Figure 6. Specifically line 267-8 states "our studies showed a marked reduction in H2AK119ub in SYO1, HS-SY-II, 1273/99 and Aska cell lines treated with TAK-981 as assessed using immunoblotting (Fig. 6C)" but panel 6C shows ChIP-Seq funnel plots instead. On the other hand there is a Figure 6E panel that is not mentioned in the results section, but which appears to show reduced H2AK119Ub levels with treatment -- although only for SYO1, not for the other three cell lines as had been stated in the text (and would be important to show, as this is another key experiment implying there is an on-target anti-synovial sarcoma mechanism of action)

Re: in vivo experiments, Figure 7: given the demonstrated effect on SS18::SSX protein in panels 5H/I, the authors' generation of FFPE blocks suitable for Ki67 IHC (panels 7G, 7O), their use of the recently-developed SS18::SSX specific antibody for CUT&RUN (line 549), the published proven value of this antibody for IHC (PMIDs: 32141887, 32559341, 35880992, 37899499), and the value of showing an on target effect in vivo, the authors should easily be able to perform additional IHC for SS18::SSX on the treated vs control mouse xenograft tumors to see if this backs up what they showed in vitro in figure 5.

****Minor comments****

Abstract: limb amputation is very rarely needed for synovial sarcoma - it would be better to justify the need for this study by saying something along the lines that current systemic therapies have little impact on survival

Stylistically a recent consensus has been reached that fusion oncogenes and their chimaeric oncoproteins are designated with a double colon (SS18::SSX) rather than a hyphen or slash as was variably done in previous literature (applies throughout manuscript, where a hyphen has been used). Also in introduction and results remember that RNA/DNA gene names should be italicized, so as to distinguish gene from protein e.g. for SS18.

Figure 2 shows SYO-1 cells for the screen experiment, but the legend and methods state HS-SY-II.

2. Significance:

Significance (Required)

This work is a tangible advance in the field, and opens the door to interesting mechanistic studies that may have implications for SUMO / epigenetic biology, as well as providing strong preclinical evidence for a new therapy -- a paper like this might mean synovial sarcomas get added to existing solid tumor trials of SUMO inhibitors. The work has direct interest to both the sarcoma foundational science community and the medical oncology community.

The science is well done overall; results are believable in the context of the existing body of literature (in the opinion of this reviewer who has expertise in the biology, pathology and medical oncology of sarcomas).

3. How much time do you estimate the authors will need to complete the suggested revisions:

Estimated time to Complete Revisions (Required)

(Decision Recommendation)

Between 3 and 6 months

4. Review Commons values the work of reviewers and encourages them to get credit for their work. Select 'Yes' below to register your reviewing activity at Web of Science

Reviewer Recognition Service (formerly Publons); note that the content of your review will not be visible on Web of Science.

No

Reviewers Comments and Rebuttal Plan, Iyer et al., 2024

Reviewer's comments (in black) and responses (in blue):

Reviewer #1 (Evidence, reproducibility and clarity (Required)):

Summary

The manuscript by Iyer and colleagues studies the exceptional dependency of synovial sarcoma (SS) cells on SUMO2. This dependency was initially identified by mining the DepMap database for selective genetic dependencies of SS cell lines. Among the combined top 200 SS-selective genes from the DEMETER and CHRONOS datasets, several members of the sumoylation machinery were identified, including SUMO2.

These findings were then first validated through two CRISPR screens in an SS cell line using sgRNAs against all these DepMap hits. The screens were performed both in vitro and in vivo in mice and 19 hits were found to be common to both screens. Of these 19 genes, two were also SS18-SSX targets, including SUMO2. Based on these findings, the effect of the sumoylation inhibitor TAK981 was tested on several SS cell lines. The drug affected cell viability with an EC50 in the nanomolar range in both 2D and 3D culture systems, while in non-SS cell lines it was considerably less potent.

RNAseq was then performed to evaluate the effect of TAK981 on the transcriptome in two cell lines. A total of 1100 genes were found to be affected, including many genes involved in cell cycle regulation and several SS18-SSX target genes. Interestingly, a doxycycline resistance signature was reversed by the treatment. A closer look at the SS18-SSX target genes revealed that TAK981 treatment led to a downregulation of SS18-SSX activated genes and to a reactivation of SS18-SSX repressed genes. To test the hypothesis that SUMO2 regulates the fusion protein itself, the effect of shRNA-mediated silencing of SUMO2 and treatment with TAK981 on its expression was assessed. Both treatments indeed led to a downregulation of SS18-SSX, suggesting that its expression is indeed under the control of SUMO2.

CUT&RUN analysis before and after treatment with TAK981 confirmed the reduction of SS18-SSX at genomic binding sites and also revealed a global reduction of H2AK119ub.

Finally, the effect of TAK981 on the growth of cell line-derived xenograft tumors was tested. TAK981 slowed the growth of Aska-SS tumors, while treatment of SYO1 tumors completely prevented its growth.

Major Comments

Figure 1

- A-B It appears that many genes are unique to one data set or the other. The overlap between the two data sets should be presented in a table or a Venn diagram.

We thank the reviewer for pointing this out. We have now added a tab to show common genes from the 2 datasets (see new tab (Common Genes) in Table.S1.)

Figure 2

B-C The x-axis of these plots is labelled 'genes' and goes up to 500. However, the sgRNA library was directed against 348 genes, while the rest are non-targeting controls. I suggest that the axis be labelled differently, e.g. sgRNA.

We appreciate the reviewer's concern. In our analysis, we targeted 348 genes with 10 sgRNAs each. Additionally, we included 183 non-targeting sgRNAs, representing about 5% of the library, with each non-targeting sgRNA treated as a separate "gene." Thus, there were 348 genes and the rest were non-targeting controls total 532 unique elements plotted on the X-axis, with the RRA score displayed on the Y-axis. We thank the reviewer for pointing out this confusion, we have now labeled the X-axis as "sgRNA".

- It is not clear why the authors restricted their selection to downstream targets of the fusion protein. This should be explained in the text.

We agree that in a selection of targets that are important for Synovial Sarcoma, one could keep the focus broad – to include any genes that are selectively essential (as found in the screen hits from our DepMap list) or limit them to a narrower list. We decided to narrow the list to include those genes that were bound by, and/or regulated by the fusion (as shown by Jereby Arnon et al.,). In our opinion this helps us focus on candidate genes that are more likely to be relevant to the fusion-driven oncogenic program. We have now provided this explanation in the main text – see line 155-157.

Figure 3

- B-D The extent of apoptosis and cell cycle arrest in case of SYO-1 cells is relatively small. The physiological effect behind the decrease in viability (Fig.3A) is therefore not very obvious in this case. Furthermore, it is stated in the text that TAK981 "cell cycle analysis using propidium iodide indicated an S-phase arrest". However, in the treated cells there are less S-phase cells detected. Taken together, it is not clear whether the effect of their drug in this case is mainly cytostatic or cytotoxic. I suggest that a cell death assay be performed, using flow cytometry to detect both apoptotic and dead cells, to determine the effect on cell viability.

We have now redone these experiments, and the flow cytometry plots and other data are now updated in the main and supplemental figures (See Supplemental Fig. S4).

- E-F These figures show essentially the same data. One of the figures could also be removed. Why are ASKA-SS cells used here and not HS-SY-II cells?

We agree that figures E and F show the same data represented using different styles – E showing an overall representation in a heatmap and F showing growth curves that help

appreciate the progressive growth changes over time and concentration. For these 2D and 3D assays, we used Aska-SS instead of HS-SY-II cells as they have been shown to form spheroids (Naka, N., Takenaka, S., Araki, N., Miwa, T., Hashimoto, N., Yoshioka, K., Joyama, S., Hamada, K.-i., Tsukamoto, Y., Tomita, Y., Ueda, T., Yoshikawa, H. and Itoh, K. (2010), Synovial Sarcoma Is a Stem Cell Malignancy^{†‡§}. *Stem Cells* 28: 1119-1131. <https://doi.org/10.1002/stem.452>). We reasoned that the ability of the Aska-SS cells to grow in spheroids would help enable robust 2D and 3D culture assays.

- G. The cells apparently do not form colonies, similar to other sarcomas. The visible clouds of cells probably originate from uneven seeding. In my opinion, this cannot be called a colony forming assay. I suggest that this panel be removed from the figure.

We appreciate the reviewer's comments – the visible clouds appear since many SySa cell lines are dislodged while washing due to their inherent loosely adherent nature. We have now repeated these assays with an alternative protocol. Please find the new figures Fig. 3F and its quantification Fig. 3G.

Figure 4

- D. The finding that the doxorubicin resistance signature is reversed by TAK981 treatment is interesting. The authors could check the combinatorial effects of TAK981 and doxorubicin experimentally.

We agree that the reversal of genes involved in doxorubicin resistance is interesting to investigate further. To address the important comment above, *we subjected 3 different SySa cell lines (HS-SY-II, 1273/99 and Yamato-SS) to a combinatorial matrix of concentrations of TAK-981 and Doxorubicin.* Importantly, all the SySa cell lines *show a strong synergistic effect* with the two drug as calculated by their Bliss score (HS-SY-II mean Bliss score of 10.2, 1273/99 mean Bliss score of 13.52 and Yamato-SS mean Bliss score of 10.78). It is to be noted here that the cell lines Yamato-SS and 1273/99 show either no inhibition with Doxorubicin or show inhibition at very high concentrations (> 10uM). but are highly sensitive to doxorubicin in the presence of TAK-981. We are thankful to the reviewer for improving our manuscript, since we agree that this is an important observation that TAK981 synergizes with Doxorubicin to augment antiproliferative effects in a panel of SySa cell lines. This is now added to the manuscript – see new Fig. 4E and Supplemental Fig. S7)

Figure 5

- G The Western blot contains vertical lines between the two lanes with shRNA-1 treated sample and the rest of the samples, indicating that these samples were run on a different gel and the blot was spliced afterwards. Although the samples with the same shRNA treatment (plus/minus Dox) were obviously run on the same blot, it is necessary to indicate the different blots. This can be done by showing the blots separately next to each other.

The blot, including all the shRNAs is a singular blot, the raw image of the blot is pasted below.

- G-I Please indicate which band(s) represent SS18-SSX. Does any of the bands correspond to the wild-type protein?

SS18-SSX1 runs as a doublet as described in the original manuscript where this antibody was developed. See Fig. 1 from Baranov et al., pasted below. It is apparent that these two bands are both specific for the fusion, and do not cross-react with the wild-type SS18 or SSX proteins as detailed in this manuscript and in many other manuscripts. To drive home this point, SSX knockdown using shRNAs, which only targets the fusion in these cells (since wildtype SSX is not expressed) leads to a reduction in both these bands, showing that both bands are specific to the fusion. This has also been shown using SS18-SSX fusion specific siRNAs. It is likely that the extra band that is still specific to the fusion represent a post-translational modification of the protein. Additionally, as evident from our new Western Blotting data from 5 SySa cell lines and HEK293-T cell controls, we see the two bands in all the 5 SySa cell lines but not in the HEK293s.

Please see supplemental figure S5.

Reference: A novel SS18-SSX fusion-specific antibody for the diagnosis of synovial sarcoma, Baranov E, McBride MJ, .. Kadoch C, Hornick JL, Am J Surg Pathol 2020

Figure 6

- F-G Tracks showing the difference between TAK981 and DMSO would show the effect of the treatment more clearly.

We have now remade these figures to show the treatment effect on both the fusion occupancy as well as the H2AK119ub more clearly (See revised Fig. 6)

- It should be evaluated whether the reduction of H2AK119ub is specific for SS18-SSX binding sites.

We thank the reviewer for pointing this out. To address this question systematically, we have redone the analysis by asking reanalyzing the effect of TAK981 on H2AK119ub a) generally, b) on SySa target genes and c) on other expression-matched genes that demonstrate H2AK119ub. See our new analysis in lines 288-294 of the text and revised Fig. 6 C-G

Figure 7

- B Some tumors in the control group are very large at the experimental endpoint (in the range of 4000 mm³). Is this consistent with their animal license?

Our AUF stipulates that mice with visible tumors with a volume equal to or lesser than 4000 mm³ will be sacrificed as in prior publications and the guidelines from the NIH. During the penultimate measurement (Day27) for the experiment, the largest tumor volume measured was 3311.28 mm³ which is in line with our AUF. However, the mice were sacrificed 2 days later to accommodate the weekend, during which the mouse with the largest tumor grew to be 4111mm³. We have now excluded the tumor volumes for all mice for that day (Day 30) to keep in accordance with our animal license. We thank the reviewer for pointing out this important consideration. As seen in the new figure, the differences in vehicle compared to drug arms are still statistically significant.

Minor comments

Figure 3

- F Concentration labelling is missing.

We have now fixed this.

Figure 4

- Text line 199-203: "Upregulated" and "downregulated" genes are mixed up.

We thank the reviewer for pointing this out. The wording of the statement was confusing, and we have now reworded it.

Figure 5

- D Is the labelling "direct targets" correct here or are these also indirect targets as in E?

The labeling of indirect targets of SySa, in SYO1 in Fig.5D is correct. Fig. 5E shows the indirect targets from the cell line HS-SY-II.

Figure 6

- In the text, the immunoblot is described as C, but the figure is actually 6E.

We thank the reviewer for pointing this out, we have remade the figure panel and the immunoblot is currently Fig. 6D

Figure 7

- B The x-axis labelling is missing and the y-axis labelling should be mm³.

This is now fixed.

- J The tumors in this experiment were very small at the start of treatment. So it is more of an engraftment assay rather than a tumor growth assay.

We injected the cells and then when the tumors were palpable, started dosing the vehicle and drug arms. Furthermore, after sacrificing, we did observe clear tumors in all arms (as evidenced by the Ki67 and the new SySa fusion stains) – except that in the drug treated mice, the tumors had smaller and less proliferating (Ki67) tumors, more characteristic of a tumor growth assay.

Reviewer #1 (Significance (Required)):

Significance

General assessment: Many fusion protein-driven sarcomas, including SS, have a poor prognosis and effective new therapies are urgently needed. However, the driver fusion proteins composed of TFs or epigenetic regulators (such as SS18-SSX) are difficult drug targets.

Therefore, upstream regulators or downstream effectors that are easier to inhibit are of great therapeutic interest. The identification of SUMO2 as a regulator of SS18-SSX expression in SS cells, which may be the reason for the selective dependence of SS cells on this gene, is represents an important advance into this direction.

Strength and limitations: Overall, the study is well conducted, the methodology is mostly appropriate, and the conclusions are reasonable and supported by evidence. An exception however are the physiological experiments, which are rather superficial and do not provide a clear picture of the physiological effects of TAK981. An important ethical criticism concerns the size of the xenograft tumors, which were grown here to 4000 mm³, exceeding the size limit normally considered in ethical guidelines. From a scientific point of view, my overall assessment of this paper is positive, but I think the above mentioned comments need to be addressed.

My own expertise is in the field of translational cancer research with a special focus on sarcoma, the mechanisms of action of oncogenic fusion proteins, generation of new tumor models and identification of novel therapeutics.

Reviewer #2 (Evidence, reproducibility and clarity (Required)):

The authors seek to evaluate genetic vulnerabilities in synovial sarcoma in an effort to identify possible new targets for therapies. Synovial sarcoma is a soft-tissue sarcoma that occurs most frequently in children and young adults. Treatment options for metastatic and recurrent diseases are limited, making the work by Iyer and colleagues timely and important. Iyer utilizes several datasets of genomic screens to identify a set of genes and gene pathways enriched in synovial sarcoma compared to other data-sets. With CRISPR screens of these identified genes, they identify SUMO2 as a top candidate hit. They explore the impact of TAK-981, a SUMO2 inhibitor, on synovial sarcoma cell line growth and on overall gene transcription, demonstrating that it specifically modifies the SS18-SSX transcriptomic program. Finally, they explore the role of TAK-981 *in vivo*.

Overall, Iyer et al demonstrate an intriguing new clinical target for synovial sarcoma therapy. To strengthen and support their analysis, we recommend the following for consideration:

- Throughout the manuscript, the authors use TAK-981 to demonstrate the impact of SUMO2 inhibition. To ensure that the effects on cell growth inhibition and cell cycle arrest are on target, it would be helpful for readers to see the experiments also completed with modulation of SUMO2 through knockdown or knockout.
- The authors utilize two synovial sarcoma cell lines implanted subcutaneously to demonstrate the impact of TAK-981 *in vivo*. They demonstrate an impact on tumor growth and overall proliferation. However, it would be helpful to the authors to know whether TAK-981 is acting on target in the *in vitro* studies.

We thank the reviewer for these comments and for suggesting to do knockdown or knockout experiments in addition to our studies on TAK981.

We have shown

- 1) genetic SUMO2 depletion in the DepMap dataset as being highly selective for synovial sarcoma cell lines (Fig. 1a),
- 2) SUMO2 as one of the highest depleted sgRNAs in our CRISPR screen both *in vitro* and *in vivo* using SUMO2 sgRNAs (see Fig. 2B and Fig. 2C).
- 3) New data showing that SUMO2 shRNA has a significant reduction of SySa cell line growth (new Fig. 3H and 3I)
- 4) As for on target activity of the TAK981, our experiments with TAK981 show stronger antiproliferative effects in SySa compared to nonSySa cel lines (See Fig. 3A) and
- 5) both SUMO2 shRNA genetic depletion in HS_SY-II (Fig. 5G) as well as in SYO1(Supplemental Fig. S9) and TAK981 treatment (Fig. 5H and Fig. 5I) downregulate the SySa fusion itself.

- In Figure 3 Panels B and C, there is no notation of what TAK-981 is used for the data provided. It is difficult to assess the results without understanding which concentration is used. Moreover, the percent cells staining with Annexin V is quite low in 3B - interpretation of this low percentage would be improved with an understanding of the dose utilized

For interpretation by the reader, it would be helpful to provide a representative scatter plot or histogram from the flow cytometry in Figure 3 panels B-D. We have now provided the concentration used for these experiments. We have also provided the representative scatter plot of the FACS data (see Supplemental Fig. S4).

- In Figures 5H-I and 6E, there are no immunoblots provided within the figure.

Figures 5H and 5I show immunoblots of HS-SY-II and SYO1 cells treated with various concentrations of TAK-981 and probed for SS18::SSX1 fusion and vinculin as loading control. We have remade Fig. 6 panel and Fig. 6D is an immunoblot of SYO1 cells with 3 replicates of DMSO or TAK-981 treated whole cell lysates probed with H2AK11ub and vinculin as loading control.

- In Figure 3F, it is difficult to assess the data due to the compressed nature of the Y axis. Additionally, the X axis does not provide units

We have now remade the figure by changing the axis. See new Fig. 3E

- In Figure 3G, the photos provided of the clonogenic assay are blurry and difficult to assess. Quantification of the experiment is not provided. Additionally, the large cluster of cells at the center of the plate likely limits the ability to identify individual colonies, as is necessary for a clonogenic assay.

We agree with the reviewer's concern – we have now repeated these assays and provided much clearer images and also quantification. See new figure (Fig. 3F&G)

- In Figure 5 panel G, the loading control is hard to appreciate for the shRNA4 +dox sample - it is unclear if it is underloaded or if it is the image quality.

We have now provided a better exposure of the blot.

- In Fig 1A-B, the authors utilize a word cloud to demonstrate the possible genes identified through their analysis. It would be helpful for authors to understand the difference in p value that is represented by the different font sizes

In Fig 1A-B, the word cloud represent the negative log₁₀ P value, and the font size is proportional to these values, we feel this is the best visual representation of >200 genes in one figure. In order to provide more quantification of this, we have now attached the actual table used which shows the values plotted directly in the word cloud so that the reader can make a more quantitative determination as well. We thank the reviewer for pointing this out. See new tabs (RNAi_WordCloud and CRISPR_WordCloud) in Supplementary Table 1

- In line 299, the second 'in' does not need to be italicized

This is now fixed.

Reviewer #2 (Significance (Required)):

- *Synovial sarcomas are lethal soft tissue sarcomas of children and young adults with very limited treatment options, especially for relapsed and metastatic disease. This paper is timely and important given that TAK-981 is already in clinical trials. The mechanistic insights are useful as well and in line with existing evidence of SS being primarily an epigenetically driven cancer. Overall, I am supportive of publication with some revisions.*

Reviewer #3 (Evidence, reproducibility and clarity (Required)):

- Authors use both a DEPmap analysis, their own screen and literature precedents to hone in on SUMO2 as a targetable dependency in synovial sarcoma, a disease with clinical need for new systemic treatments. A new SUMO targeting drug (TAK-981) is then shown to be effective against multiple synovial sarcoma cell lines (and cell line xenografts), in terms of growth/apoptosis assays, reversing disease-specific oncogenic transcription patterns, and decreasing levels of the key synovial sarcoma oncoprotein SS18::SSX and its H2AK119Ub partner in terms of total protein and chromatin occupancy (including at target genes).

Major comments:

- The supplemental tables S1, S2, S3 and S4 were not included in the materials.

We apologize this happened during the upload stage since they were separate PDFs that failed to upload for some reason, these were later provided during the review process.

- Related re: Fig 1: The Analysis of functional genomic screens does not provide details on what cell lines were included -- in particular which and how many synovial sarcoma cell lines. The supplementary tables which may have had this information were not included with the submitted material nor can they be found on the bioRxiv posted version, but even if provided in supplemental this information should be brought into the main text of methods and/or the first part of the results. It is important that screening results for "synovial sarcoma" are not based on data from the notorious (in the synovial sarcoma field) SW982 cell line. SW982 carries a BRAF mutation characteristic of melanoma and lacks the SS18::SS18 oncogene, and so despite its persistent ATCC annotation as derived in the 1970s from a "synovial sarcoma", this cell line is definitely not a synovial sarcoma. If it has been included, then the analysis has to be redone excluding it and Figure 1 has to be re-rendered.

The DepMap RNAi and CRISPR databases that we used for our analyses has data from cell lines FUJI, HS-SY-II, 1273/99 and SS1A but not from the cell line SW982. Therefore our analyses are not affected. It would have affected our data when we compared gene expression of synovial

and non synovial cell lines as in Fig. 1 e. but since in this figure too, we were only focusing on cell lines where there was screening data (not just expression data), SW982 was fortunately not present. All the cell lines used in our studies – HS-SY-II, SYO1, Aska-SS, Yamato SS, and 1273/99 are SySa fusion positive cell lines and in addition to STR profiling, we have also now performed a fusion-specific antibody immunoblot on all the cell lines to confirm that they express the SySa fusions. See new supplemental Fig.S5

- Re: Fig 3: while the cell lines are described as STR authenticated, it is important that the four synovial cell lines be specifically confirmed to express SS18::SSX (and conversely that the control cell lines do not express this). For these experiments, the number of cell lines and the use of two non SySa cell lines is good. However, in terms of controls for the drug testing, while the DMSO negative control is important, for context -- in a manuscript claiming to be identifying a new treatment for synovial sarcoma -- there should also be a positive control drug treatment arm - i.e. a comparison of at least some of the assay results shown in this figure to results obtained in the same cell line(s) with a drug currently used for SySa treatment ... like doxorubicin (or trabectedin, or pazopanib). Indeed, based on the penultimate sentence in the discussion, a combo arm of TAK + doxorubicin might also be of interest.

We thank the reviewer for the comment. As mentioned above, we had previously tested and now retested the HS-SY-II, SYO1, Aska-SS, Yamato SS, and 1273/99 using the SySa fusion specific antibody see fig. 5H and 5I as well as in supplemental figure S9 and now new supplementary Fig. S5.

As for the doxorubicin question, we have now performed experiments treating three SySa cell lines HS-SY-II, 1273/99 and Yamato-SS with doxorubicin and TAK981. Interestingly, our results show that even as these cell lines are mostly insensitive to doxorubicin by themselves, addition of TAK981 shows potent synergistic cell growth reduction (see new Fig. 4E and Supplemental Fig. S7). This is consistent with our transcriptomic data showing a very strong reduction in Doxorubicin resistance genes upon TAK981 treatment.

- Re: Fig 5. This is a great way to address if TAK-981 is affecting SS18::SSX induced biology. My concern is that perhaps the best supporting data has been cherry-picked and selectively presented. Specifically, the full list of Jerby-Arnon "core oncogenic program" genes should be shown so the reader can assess not only the ones where the expression is reversed, but also for how many of those genes there is actually no effect.

We have now shown heatmaps for all the genes in the Jerby-Arnon “core oncogenic program” in the new Supplemental Fig. S6 (A-D) In these new figures, it can be appreciated that a very large proportion of the SySa target genes are reversed by TAK-981 treatment (as evidenced by the highly statistically significant GSEA normalized enrichment scores) in addition to the leading edge genes displayed in the main Figure.

- The key finding (Fig 5G) that SUMO2 knockdown leads to a decreased level of the SS18::SSX oncoprotein itself is only shown for HS-SY-II cells; this particular finding is especially important to assess in multiple cell lines (as is done for many other less-critical figure panels in this manuscript).

We thank the reviewer for pointing this out. We have now knocked down SUMO2 in another SySa cell line SYO1 and see similar reduction in the fusion. See Supplemental Fig. S9

- Some of the text in the results section "SUMO2 inhibition diminishes SS18-SSX chromatin occupancy..." describing Figure 6 does not match what is actually presented in Figure 6. Specifically line 267-8 states "our studies showed a marked reduction in H2AK119ub in SYO1, HS-SY-II, 1273/99 and Aska cell lines treated with TAK-981 as assessed using immunoblotting (Fig. 6C)" but panel 6C shows CHIP-Seq funnel plots instead. On the other hand there is a Figure 6E panel that is not mentioned in the results section, but which appears to show reduced H2AK119Ub levels with treatment -- although only for SYO1, not for the other three cell lines as had been stated in the text (and would be important to show, as this is another key experiment implying there is an on-target anti-synovial sarcoma mechanism of action)

We thank the reviewer for pointing this discrepancy out, which we have now fixed. We have done a more extensive analysis of H2AK119ub, which shows 1) a generally higher H2AK119ub on SySa target genes compared to expression matched controls (see new Fig.6E), and a higher reduction of H2AK119ub on SySa target genes upon TAK981 treatment (see new Fig.6F) compared to nontarget genes (see new Fig. 6G). In addition, addressing the last part of the comment, we have now added H2AK119ub data in 2 other cell lines – Yamato SS and 1273/99 - where the result is consistent – TAK981 treatment leads to a reduction in H2AK119ub. See new supplemental figure S12.

Re: in vivo experiments, Figure 7: given the demonstrated effect on SS18::SSX protein in panels 5H/I, the authors' generation of FFPE blocks suitable for Ki67 IHC (panels 7G, 7O), their use of the recently-developed SS18::SSX specific antibody for CUT&RUN (line 549), the published proven value of this antibody for IHC (PMIDs: 32141887, 32559341, 35880992, 37899499), and the value of showing an on target effect in vivo, the authors should easily be able to perform additional IHC for SS18::SSX on the treated vs control mouse xenograft tumors to see if this backs up what they showed in vitro in figure 5.

We agree this is important – we have now performed IHC staining on in vivo tumor sections with the SS18-SSX fusion antibody and our results show that just like in our in vitro studies, TAK981 treated tumors show a significantly diminished SS18-SSX fusion. Please see our new IHC data from SYO1 implanted mice in Fig.7S-R.

Minor comments

Abstract: limb amputation is very rarely needed for synovial sarcoma - it would be better to justify the need for this study by saying something along the lines that current systemic therapies have little impact on survival.

We have now removed this comment and would like to thank the reviewer for pointing this out.

Stylistically a recent consensus has been reached that fusion oncogenes and their chimaeric oncoproteins are designated with a double colon (SS18::SSX) rather than a hyphen or slash as was variably done in previous literature (applies throughout manuscript, where a hyphen has been used). Also in introduction and results remember that RNA/DNA gene names should be italicized, so as to distinguish gene from protein e.g. for SS18.

We have now made this change throughout the revised manuscript.

Figure 2 shows SYO-1 cells for the screen experiment, but the legend and methods state HS-SY-II.

We have now fixed this.

Reviewer #3 (Significance (Required)):

Significance:

This work is a tangible advance in the field, and opens the door to interesting mechanistic studies that may have implications for SUMO / epigenetic biology, as well as providing strong preclinical evidence for a new therapy -- a paper like this might mean synovial sarcomas get added to existing solid tumor trials of SUMO inhibitors. The work has direct interest to both the sarcoma foundational science community and the medical oncology community. The science is well done overall; results are believable in the context of the existing body of literature (in the opinion of this reviewer who has expertise in the biology, pathology and medical oncology of sarcomas).

We thank the reviewers for their comprehensive review of our manuscript substantially improving the revised version.

Dear Dr Deshpande,

Thank you for submitting your revised manuscript (EMBOJ-2025-120758-T) to The EMBO Journal, as well for your patience with our response. Your amended study was sent back to the three referees for their scientific reassessment, and we have received detailed re-reports from two of them, which I enclose below. Please note that while referee #1 was unfortunately not able to reassess your work, we have now editorially evaluated your response to this expert and found the issues raised to be addressed satisfactorily. As you will see, the other experts state that the work has been substantially enhanced by the revisions and they are now broadly in favour of publication, pending minor revision.

Thus, we are pleased to inform you that your manuscript has been accepted in principle for publication in The EMBO Journal.

Please carefully consider the remaining minor points raised by referee #3, revising data presentation and annotation.

Also, we now need you to take care of a number of minor issues related to formatting and data annotation, which I will share shortly in a separate message, together with additional changes and requests by our production team including for Source Data provision.

Please contact me at any time if you have additional questions related.

As you might have noted from our webpage, every paper at the EMBO Journal now includes a 'Synopsis', displayed on the html and freely accessible to all readers. The synopsis includes a 'model' figure as well as 2-5 one-short-sentence bullet points that summarize the article. I would appreciate if you could provide this figure and the bullet points.

Thank you for giving us the chance to consider your manuscript for The EMBO Journal. I look forward to your final revision.

Again, please contact me at any time if you need any help or have further questions.

Best regards,

Daniel Klimmeck

Referee #2:

The authors have addressed my previous concerns adequately and I have no further comments.

Referee #3:

The comments and critiques in my original Review Commons report have all been adequately addressed, including by generating recommended new supporting data (which does further support the authors' original conclusions).

My additional comments are all quite minor:

- the .pdf versions of Supplemental Tables S1, S3 and S4 do not properly render the contents of the .xls versions (at least on the review site)
- response to the Re: Fig 3 critique points to "new Fig. 5E" but the relevant data appears to be in Fig. 4E
- response to comment about Fig. 5g points to Supplemental Fig. 9 but the relevant data appears to be in Supplemental Fig. 10
- response to comment about Fig. 6 points to "new Supplemental Fig. S11" but the relevant data appears to be in Supplemental Fig. S12

Rev_Com_number: RC-2024-02710

New_manu_number: EMBOJ-2025-120758-T

Corr_author: Deshpande

Title: SUMO2 Inhibition Reverses Aberrant Epigenetic Rewiring in Synovial Sarcoma and Impairs Tumorigenesis

Dear Dr Deshpande,

Further to below, please find enclosed the mentioned additional formatting requirements for your article.

Please let me know any time of you have additional questions related.

We look forward to your final resubmission, mission.

Best regards,

Daniel Klimmeck

>> Please provide the main manuscript text as .docx file.

>> Author Contributions: Remove the author contributions information from the manuscript text. Note that CRediT has replaced the traditional author contributions section as of now because it offers a systematic machine-readable author contributions format that allows for more effective research assessment. and use the free text boxes beneath each contributing author's name to add specific details on the author's contribution.

More information is available in our guide to authors.
<https://www.embopress.org/page/journal/14602075/authorguide>

>> Adjust the title of the 'Conflict-of-interest disclosure' section to 'Disclosure and Competing Interests Statement'.

>> Provide a completed Author Checklist.

>> Add complete annotation of animal husbandry - mouse ethics to the Methods and adjust the Author Checklist accordingly.

>> Section order should be corrected as follows: title page with complete author information, abstract, keywords, introduction, results, discussion, methods, data availability section, acknowledgements, disclosure and competing interests statement, references, main figure legends, tables, expanded figure legends.

The figure legends should be compiled at the end of the manuscript text

>> The "Significance" paragraph should be removed from the manuscript text.

>> Figure callouts: Please ensure that the following figures and panels are called out in sequential order: Fig.3B, 3C, 5F.

>> Please provide source data for the study as to the separate request e-mail.

>> References: please adjust reference format to EMBO Journal format, 10 authors et al, and place References after the Discussion, before figure legends.

>> Please recheck references for the bioRxiv entries Dempster et al. (2019) & Love et al. (2014) and update the citations if in the meantime published as regular article.

>> Please add a Reagents and Tools table to the Methods section, as a separate file using the existing template in the Guide For Authors, listing key reagents, experimental models, software and relevant equipment.

>> Dataset EV legends: Table S1 and Table S4 should be renamed "Dataset E1" and "Dataset EV2". Tables S2 and S3 should be renamed "Table EV1" and "Table EV2". All need legends added to the files.

>> Appendix: add a table of contents (with page numbers) to the first page. The nomenclature needs correcting to "Appendix Figure S1" etc, and the legends should be moved underneath each corresponding figure.

>> Biorender: remove from Acknowledgements and add a section to the Data Availability Section, using the following format:
Graphics:
(some of the... OR Figure #... OR synopsis) Graphics were created with BioRender.com.

>> Data availability section: please add a specific URL for GSE276074 dataset.

>> Please indicate redisplay of data from:

>>>> Appendix figure S9 in the figure legend of Appendix Figure S11.

>>>> Appendix figure S12 in the figure legend of Appendix Figure S10.

>> Please provide higher-resolution images for

>>>> Appendix Figures S1B / S2B / S5A&B / S7A.

>> Consider additional changes and comments from our production team as indicated below:

- Figure legends:

1. Please note that the exact p values are not provided in the legends of figures 3B, C, D, G, I; 7C, J, K, N, P, Q, S
2. Please indicate the statistical test used for data analysis in the legends of figures 1D; 3B, C, D, G, I; 4A, B, E
3. Please indicate what * / ** / *** / **** represents; if this represents p value(s), please indicate the statistical test used and where appropriate, specify the exact p value in the legend(s) of figure(s) 7B, F, H, I
4. Please note that information related to n is missing in the legends of figures 3E, 4A, B; 7B, N, P, Q
5. Although 'n' is provided, please describe the nature of entity for 'n' in the legends of figures 3A, B, C, D, G, I; 7C, J, K
6. Please note that the error bars are not defined in the legends of figures 7B, J, K, N, P, Q, S
7. Please note that the measure of center for the error bars needs to be defined in the legends of figures 3A, B, C, D, E, G, I; 7C
8. Please note that scale bar and its definition are missing for figures 7E, G, M, O

The authors addressed the remaining editorial issues.

Dear Dr Deshpande,

Thank you for submitting the revised version of your manuscript. I have now evaluated your amended manuscript and concluded that the remaining minor concerns have been sufficiently addressed.

I am thus pleased to inform you that your manuscript has been accepted for publication in the EMBO Journal.

On a different note, I would like to alert you that EMBO Press offers a format for a video-synopsis of work published with us, which essentially is a short, author-generated film explaining the core findings in hand drawings, and, as we believe, can be very useful to increase visibility of the work. Please see the following link for representative examples and their integration into the article web page:

<https://www.embopress.org/doi/full/10.15252/emj.2019103932>

Best regards,

Daniel Klimmeck

Daniel Klimmeck, PhD
Senior Editor
The EMBO Journal
EMBO
Postfach 1022-40
Meyerohofstrasse 1
D-69117 Heidelberg
contact@embojournal.org

Rev_Com_number: RC-2024-02710

New_manu_number: EMBOJ-2025-120758R

Corr_author: Deshpande

Title: Targeting SUMO2 Reverses Aberrant Epigenetic Rewiring Driven by SS18::SSX Fusion Oncoproteins and Impairs Sarcomagenesis